# Role of immigrant males and muzzle contacts in the uptake of a novel food by wild vervet monkeys

**Pooja Dongre[1,2], Gaëlle Lanté[1,3], Mathieu Cantat[2], Charlotte Canteloup[1,2,4†], Erica van de Waal[1,2*†]**

[1]Department of Ecology and Evolution, University of Lausanne, Lausanne, Switzerland; [2]Inkawu Vervet Project, Mawana Game Reserve, KwaZulu Natal, South Africa; [3]University of Poitiers, Poitiers, France; [4]Laboratory of Cognitive & Adaptive Neurosciences, CNRS - UMR 7364, University of Strasbourg, Strasbourg, France

**Abstract** The entry into and uptake of information in social groups is critical for behavioral adaptation by long-lived species in rapidly changing environments. We exposed five groups of wild vervet monkeys to a novel food to investigate the innovation of processing and consuming it. We report that immigrant males innovated in two groups, and an infant innovated in one group. In two other groups, immigrant males imported the innovation from their previous groups. We compared uptake between groups with respect to the initial innovator to examine the extent to which dispersing males could introduce an innovation into groups. Uptake of the novel food was faster in groups where immigrant males ate first rather than the infants. Younger individuals were more likely overall, and faster, to subsequently acquire the novel food. We also investigated the role of muzzle contact behavior in information seeking around the novel food. Muzzle contacts decreased in frequency over repeated exposures to the novel food. Muzzle contacts were initiated the most by naïve individuals, high rankers, and juveniles; and were targeted most towards knowledgeable individuals and high rankers, and the least towards infants. We highlight the potential importance of dispersers in rapidly exploiting novel resources among populations.

***For correspondence:**
erica.vandewaal@unil.ch

†Joint last authorships

**Competing interest:** The authors declare that no competing interests exist.

## Editor's evaluation

This important study provides new insights into behavioural mechanisms involved in the transmission of information surrounding innovation in a social species. Combining experimental and observational evidence, the results are solid and convincing regarding the effects of age, rank and muzzle contacts in transmitting knowledge among vervet monkeys. The work will be of interest to ethologists, behavioural ecologists and comparative psychologists.

## Introduction

To thrive in rapidly changing environments, including those induced by humans, animals must respond quickly to relevant information about their surroundings (*Barrett et al., 2019*). Climate change or human-induced invasions of species novel to the area, as well as the introduction of human artifacts into the environment can affect different species in myriad ways, for example, bringing new threats, disruptions, competition, or novel resource opportunities. Adaptive behavioral responses to such changes can include effectively avoiding new predators, maintaining the high competitive ability, and exploiting novel resources (*Barrett et al., 2019*; *Jesmer et al., 2018*; *Gruber et al., 2019*; *Sih et al., 2011*). For long-lived species, fast, learned behavioral adaptations are crucial for survival when

circumstances change too rapidly for genetic adaptation to suffice. Whilst transmission mechanisms of genetic adaptation are well understood, our understanding of how behavioural adaptations arise and spread is murky, and the role of individual heterogeneity in a group remains underexplored (*Jolles et al., 2020*).

Research has identified two main classes of behavioral responses to novel stimuli in animals. These are neophobia and exploration (*Forss et al., 2017*; *Carter et al., 2012*). Neophobia refers to the avoidance of novelty, which could expose individuals to risky situations such as entering unknown home ranges of predators or ingesting toxins. Neophobia is common in response to potential novel foods (*Modlinska and Pisula, 2018*). Exploration, on the other hand, involves behaviors that seek information about novel stimuli. Obtaining novel information directly from the environment requires overcoming neophobia and engaging in exploration, tendencies for which may vary between individuals (*Jolles et al., 2020*). If this information is used, it can facilitate innovation. (*Kummer and Goodall, 1985*) defined innovation as, 'a solution to a novel problem, or a novel solution to an old one,' and 'a new ecological discovery such as a food item not previously part of the group.' Behavioral innovations can, therefore, allow species with slow generational turnover to adapt their behavior quickly to changing circumstances, for example, to exploit a novel resource introduced into the current habitat (e.g. *McLennan and Hockings, 2014*). To innovate, however, it is necessary to go beyond obtaining information through exploration. Individuals must interact with the environment in novel ways, either using novel behaviors with known environmental features, or performing familiar behaviours on novel aspects of the environment, which additionally requires behavioural plasticity (*Brosnan and Hopper, 2014*). This plasticity can also be highly variable, both between individuals of a species, and within individuals across time (*Modlinska and Pisula, 2018*). Given the risks associated with novelty and innovation, it is likely only beneficial to innovate when necessary; and motivation based on the physiological states of individuals is likely important in variation in innovation within species (*Brosnan and Hopper, 2014*; *Sol, 2015*). Moreover, neophobia and motivation to innovate may vary over time according to an individual's current needs, developmental status, and transient environmental conditions (*Modlinska and Pisula, 2018*; *Sol, 2015*). For example, innovation might be more likely in juveniles who may be less neophobic due to their need to learn about their environment before adulthood, or in dispersing individuals who need to find a new home territory. Accordingly, dispersing individuals may go through a transitory exploratory behavioral syndrome (*Sih et al., 2012*) at the time of dispersal making them more likely to innovate during that period. Greater behavioral flexibility ('… adaptive change in the behavior of an animal, in response to changes in the external or internal environment…,' including stopping or starting a behavior *Brown and Tait, 2014*) is an important requisite for innovation and is apparent in both juvenile (*Kumpan et al., 2020*) and dispersing male (*Bono et al., 2018*) vervet monkeys.

On the other hand, if innovative conspecifics or individuals that uniquely possess particular knowledge are present, individuals can save energy and avoid risks by learning socially from them. Indeed, many studies of diverse species in captivity have found that observing a conspecific eating a novel food reduces neophobic responses (*Forss et al., 2017*; *Modlinska and Pisula, 2018*). A study on wild jackdaws found the same (*Greggor et al., 2016*), and similarly, wild baboons with a 'bold' personality handled a novel food for longer after seeing a demonstrator do so (though 'shy' baboons did not) (*Carter et al., 2014*). Further investigation is required in the wild, since the risk for foraging animals to ingest toxins via unknown foods can be high, whilst this risk is diminished in captivity due to human feeding with edible food only. Indeed, captivity and increased exposure to human artifacts appear to increase exploration of novel objects in vervet monkeys (*Forss et al., 2022*), suggesting differences in risk-taking due to familiarity with human provisions. In addition, individual differences, such as age or sex, of the demonstrating conspecifics may be important (*Kendal et al., 2018*) and more work is needed on the topic. Moreover, in several species, dispersing individuals have been hypothesized to import information or behavioral innovations, upon immigration, into new groups (*O'Malley et al., 2012*; *Biro et al., 2003*; *Barrett et al., 2017*; *Perry, 2003*; *Péter et al., 2022*; *McDougall et al., 2010*; *Gunst et al., 2007*). For example, one study reports an immigrant vervet monkey providing spatial knowledge to his new group of a remaining water hole, during a drought, in a neighboring territory (*McDougall et al., 2010*). Detailed work in capuchins suggests the involvement of immigrants in both creating and spreading innovations in social and foraging domains (*Barrett et al., 2017*; *Perry, 2003*). Male Japanese macaques have been suspected to transfer stone handling patterns between troops

(*Gunst et al., 2007*). While these studies show that dispersing individuals might facilitate the spread of information at the population level, experimental evidence focusing on multiple groups is sparse, resulting in a small pool of evidence.

Within wild groups, animals can use social information to guide foraging decisions. Many social learning studies in primates have focused on visual access to information (e.g. *Bono et al., 2018*; *Carter et al., 2014*; *Barrett et al., 2017*; *Canteloup et al., 2021*; *van de Waal et al., 2010*; *Grampp et al., 2019*, and see review in *Whiten, 2019*). However, for Cercopithecoid monkeys, detailed olfactory information, in a foraging context, may also be acquired through muzzle contact behavior – the act of one individual bringing their muzzle into very close proximity with another's (*Laidre, 2009*; *Drapier et al., 2002*; *Chauvin and Thierry, 2005*; *Nord et al., 2021*). Indeed, previous studies found that, whilst foraging, muzzle contacts were most commonly initiated by infants and juveniles towards adults (*Nord et al., 2021*; *Lycett and Henzi, 1992*), which supports their function in information acquisition as young animals are still learning about their dietary repertoire, and adults are likely the most reliable sources of information. (*Nord et al., 2021*) also suggest that, due to the necessary close proximity, social tolerance may constrain information transmission in this modality, which can be affected by age, sex, and rank. In the presence of novel resources, muzzle contacts may be useful to adults as well as youngsters. Experimental research into this mode of information transmission in the presence of a novel resource is now required.

Vervet monkeys (*Chlorocebus pygerythrus*) are a species that thrive in natural, urban, and agricultural habitats, and are widely distributed throughout eastern sub-Saharan Africa (*Turner et al., 2019*). Their diverse habitats, including those highly modified by humans, make them an ideal species in which to investigate adaptation to novel environmental conditions. They live in multi-male multi-female troops, with philopatric females, and males dispersing multiple times during their

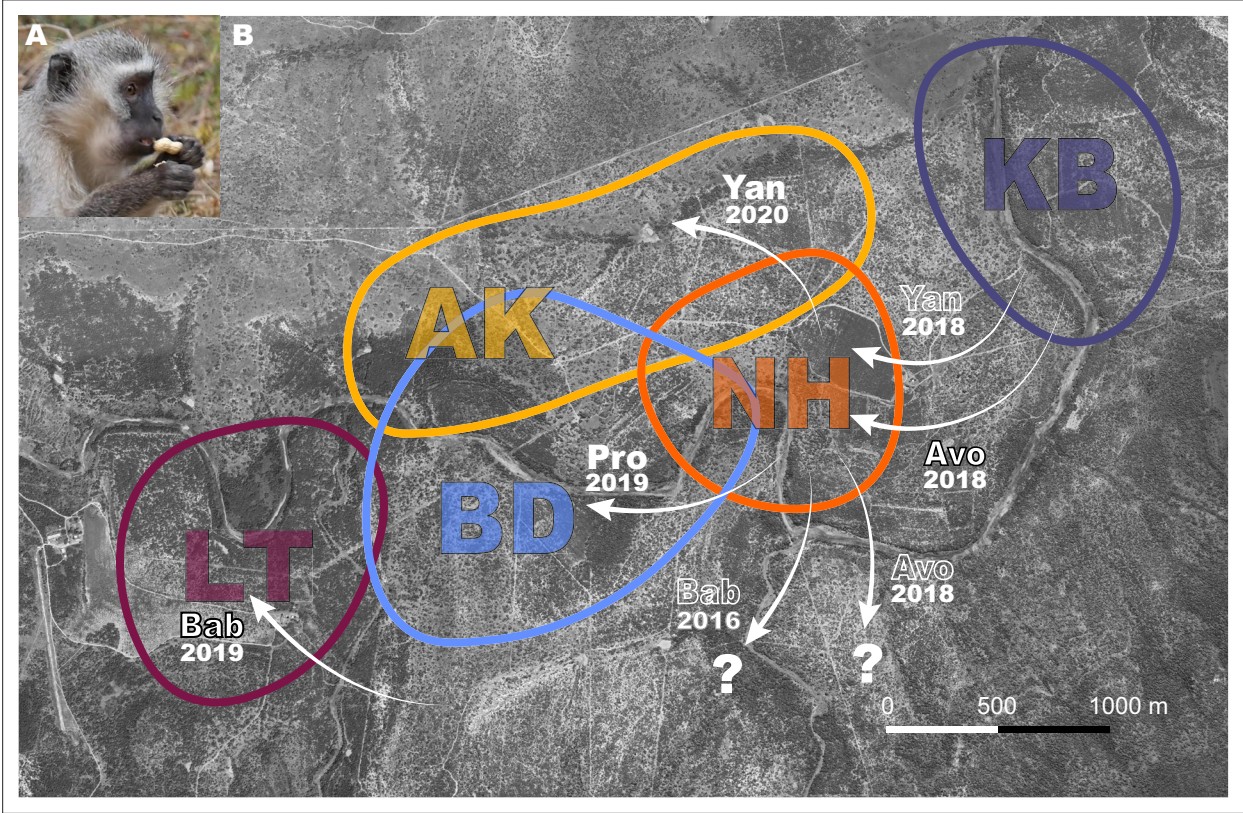

**Figure 1.** Dispersal events between study groups where monkeys were exposed to peanuts. (**A**) A vervet monkey holding an unshelled peanut, about to open it. (**B**) Aerial view of the study area with colored shapes showing a rough estimate of group home ranges for study groups AK, BD, KB, LT, and NH. White arrows with annotations represent relevant dispersals. Names of males and year of dispersal are shown. Black outlined text indicates the immigrant innovators who were naïve to peanuts, solid white text shows the immigrants that imported innovations, and white outlined text shows: parallel dispersal with innovator (*Yan* 2018); that the innovator was habituated in a study group prior to participation in this experiment (*Bab* 2016); or that the innovator left the study group (*Avo* 2018). Question mark shows that males dispersed to an unstudied group.

lives. Frequently dispersing males may serve as vectors of information between groups. In addition, if dispersal triggers an increase in exploratory behavior, necessary to seek novelty in order to leave one group to join a new one, around the dispersal period they may also become more likely innovators in novel environments (*Sol, 2015*), potentially facilitating behavioral adaptation to diverse habitats across their geographical range (*Turner et al., 2019*). In a previous study by our team (*Canteloup et al., 2021*), in 2018, two groups (NH, KB) of wild vervet monkeys were provided with a novel food that required extraction (peanuts in shells, *Figure 1A*) before consumption. The aim of this initial study was to test whether vervet monkeys socially learned how to extract peanuts from their shells, and from whom they did learn. The results supported social transmission of the opening techniques used to extract peanuts, based on visual attention to demonstrators, and that vervet monkeys socially learned the technique that yielded the highest observed payoff and that was demonstrated by higher-ranked individuals (*Canteloup et al., 2021*). Here, we replicated the same experimental paradigm in 2019 and 2020 in three more groups (AK, BD, LT) after some males from the initially studied groups dispersed to other studied groups. The aim was to investigate whether dispersing males could trigger the uptake of an innovation in their new groups. Specifically, we took advantage of the natural dispersals of males from groups already accustomed to extracting and eating peanuts (*Canteloup et al., 2021*) into groups that never had. This endeavor afforded us the opportunity to also observe innovation, which subsequently inspired hypotheses about the potential role of dispersal in innovation, building upon the work of others (*Brosnan and Hopper, 2014*; *Sol, 2015*). Our observations of innovation are limited in number, but further testing of the hypotheses we propose as a result of our exploratory analysis that we present may aid our understanding of animal innovation.

The present study addressed the following questions: First, (1a) Who innovated and how did it affect the extent to which the innovation was adopted by the group? We expected, when the innovators or initiators (in case of immigrant males importing the innovation) were adults rather than juveniles or infants, greater neophobia reduction and, therefore, faster and more widespread uptake of the novel food, in agreement with the three phases in the ontogeny of social learning in primates (*Whiten and van de Waal, 2018*). Next, to further our understanding of the uptake of innovations, we assessed (1b) whether socio-demographic characteristics (age, sex, rank) of group members predicted their adoption of the innovation at the first exposure, and across all four exposures after the first eating event. We expected adoption of the innovation to be more likely in younger monkeys due to previous findings that juveniles take more risks (*Fairbanks, 1993*), are less neophobic (*Benson-Amram et al., 2013*; *Bergman and Kitchen, 2009*; *Miller et al., 2015*; *Sherratt and Morand-Ferron, 2018*), and generally tend to learn faster (*Kumpan et al., 2020*) than adults.

Second, we experimentally investigated the function of muzzle contact behavior in novel food information acquisition. Specifically, we tested (2a) the effects of the amount of exposure to the food (and, therefore, familiarity with it) and the number of monkeys eating it on the rate of muzzle contacts. We expected that the rate of muzzle contacts would decrease as monkeys had more exposure to peanuts, if there were many monkeys eating. This would provide evidence for the function of muzzle contact in obtaining novel food information, by showing the dependence of muzzle contact rate on individuals developing their own knowledge of it through more group exposures and through eating the food. We also analyzed (2b) whether individuals' knowledge of the food, and their age, sex, and rank predicted initiating and being targets of muzzle contacts. We expected an effect of knowledge, specifically for naïve monkeys to initiate more, and knowledgeable monkeys to be targeted more, with muzzle contact being a medium to obtain information about what conspecifics are eating. When referring to knowledge we are referring to the individuals' own knowledge acquired in this specific case that peanuts are a food source. We do not invoke any more complex cognitive skills such as inference of the knowledge of others beyond what is observable of a conspecific eating the novel food. As in *Nord et al., 2021*, we also expected effects of age, with juveniles more likely initiators and adults more likely targets, as these are the theoretically predicted directions of social information transfer (*Whiten and van de Waal, 2018*), under the rationale that adults should have the most reliable information. Given the close proximity required to initiate muzzle contacts, we also expected low-ranked individuals to be less likely to initiate muzzle contacts, as they are tolerated by fewer group members (*Nord et al., 2021*).

**Table 1.** Cumulative numbers of monkeys eating at each exposure in each group, with a total of 81 (of a possible 164) monkeys eating across the whole experiment.

| Group | Group size | 1 | 2 | 3 | 4 | 5 | 6 | Total |
|-------|-----------|-----|------|------|------|------|-----|-------|
| | | **Exposure number** | | | | | | |
| AK | ~20 | 0* | 5** | 13** | 17** | 19** | - | 19 |
| BD | 65 | 19 | 25 | 29 | 32 | - | - | 32 |
| KB | 19 | 0 | 0 | 1 | 3 | 3 | 3 | 3 |
| LT | 25 | 5 | 13 | 16 | 21 | - | - | 21 |
| NH | 35 | 3 | 3 | 3 | 6 | - | - | 6 |
| | | | | | | Grand total = | | 81 |

NB. *AK in 2019; **AK in 2020.

## Results

We refer to the group AK differentially as $AK_{19}$ and $AK_{20}$, representing their status in 2019 and 2020, respectively, as 40% of the group composition changed between years due to dispersals, deaths, and changes in age categories (see *Supplementary file 1a* and detailed description in Materials and methods).

Across the experiment, a total of 81/164 vervet monkeys in all five groups, learned to successfully extract and eat peanuts during four exposures from each group's first eating event (*Table 1*).

When presented with the novel food, multiple individuals (2–16 individuals) in all groups (except BD where the knowledgeable immigrant approached the box first and immediately started eating) approached the box, looked at the peanuts, and retreated without touching any (visual inspection; *Table 2*); and at least one group member (1–7 individuals) approached and handled, sniffed or nibbled the peanuts before rejecting them and retreating from the box (contact inspection; *Table 2*).

### Who innovated and how did it affect the extent to which the innovation was adopted by the group?

When tested in 2018, in NH, a naïve immigrant male, *Avo*, was the third monkey to approach the box, and innovated extracting and eating peanuts during the group's first exposure to the novel food. In KB a naïve infant male, *Aar*, was the 16th monkey to approach the box (across all the exposures) and innovated at the group's third exposure.

In 2019, in BD, a knowledgeable immigrant male, *Pro* (who emigrated from NH, *Figure 1B*), was the first to approach the box and started extracting and eating immediately at the first exposure. In LT, an immigrant male, *Bab*, was the 17th monkey to approach the box and innovate during the group's first exposure. In AK, no monkeys innovated in 2019 ($AK_{19}$), but in 2020 ($AK_{20}$), at the group's second exposure (but the first with a knowledgeable immigrant), *Yan* (also emigrating from NH; *Figure 1B*), was the fourth to approach the box and the first to extract and eat peanuts (*Supplementary file 1a*).

During the first exposures to peanuts in BD, LT and NH, when new immigrant males initiated eating peanuts, we observed that the following percentages of these groups started to extract and eat peanuts during that exposure: BD: 31% (group n=65); LT: 20% (group n=25); NH: 9% (group n=35; *Figure 2A*). In the first exposures in $AK_{19}$ and KB, no monkeys started to extract and eat peanuts. After an immigrant male ate in $AK_{20}$, at the second exposure in that group, 30% (group n=20) of the group followed during that exposure

**Table 2.** Number of individuals in each group that showed each type of response to the peanuts before the innovator or knowledgeable immigrant started eating.

| Group | Approach box and leave | Contact exploration and rejection |
|-------|------------------------|------------------------------------|
| AK19 | 12 | 5 |
| AK20 | 3 | 1 |
| BD | 0 | 0 |
| KB | 15 | 4 |
| LT | 16 | 7 |
| NH | 2 | 1 |

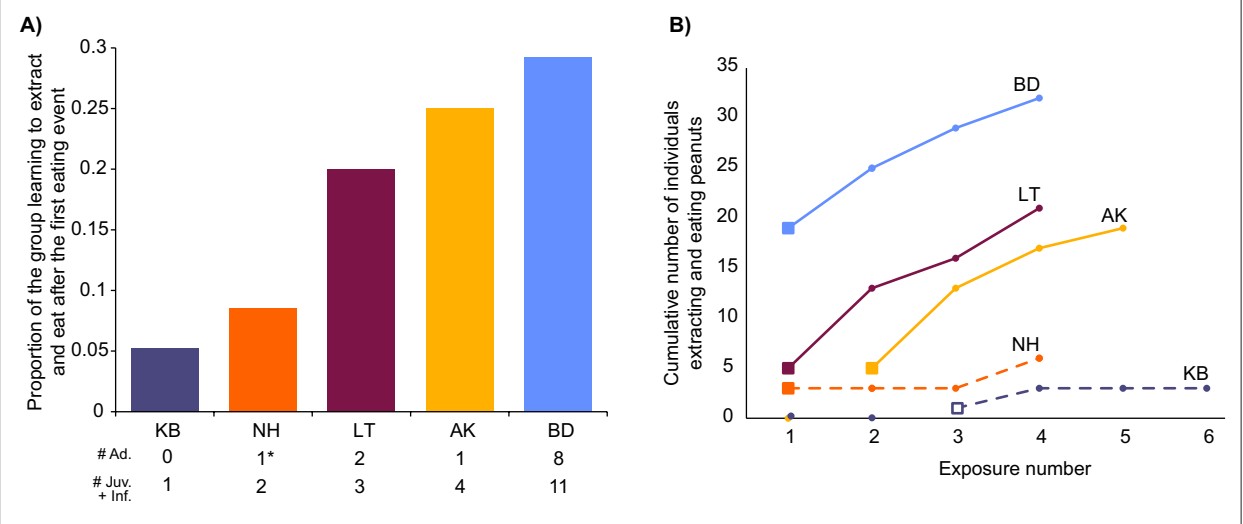

**Figure 2.** Uptake of extracting and eating peanuts in each group. (**A**) Shows the proportion of each group that started eating when the first eating event took place. Total numbers of individuals, split by age, are shown below the x-axis. In NH the asterisk highlights the one adult which was the innovator male that left after their first exposure, and was only followed by juveniles learning to extract and eat. (**B**) Shows the progression in each group over four exposures from the first eating event. Solid squares represent when immigrant males were the first to eat, and the open square shows when the infant was first to eat. Solid lines show when there were adults present who had started to eat, whereas dashed lines show when there were only juveniles and infants present that had already started to eat. Males who were knowledgeable and imported innovations from other groups (*Pro* in BD and *Yan* in AK) are excluded from totals in both panels (visualized in Microsoft Excel).

(*Figure 2A*). When an infant innovated at the third exposure in KB, no other group members followed during that exposure (*Figure 2A*). The immigrant who innovated in NH left the group after the group's first exposure, leaving just two juveniles who had also started eating at the first exposure. After four exposures from the first eating event in all groups, the percentages of each group extracting and eating peanuts were: 95% in $AK_{20}$; 66% in BD; 21% in KB; 84% in LT; and 20% in NH (*Figure 2B*).

### Socio-demographic variation in uptake of the innovation

Here, we examined whether age, sex, and rank predicted successfully extracting and eating peanuts in the first exposure with an eating event and over the four subsequent exposures. During the first exposure an eating event, we found a significant main effect of age, with juveniles 4.17 times (417%) more likely to extract and eat peanuts than adults, and 5.43 times (543%) more likely than infants, but there was no significant difference between infants and adults (Model 1, *Table 3*). We did not find any significant effect of rank and sex on extracting and eating peanuts at the first exposure with an eating event. Conditional $R^2$=0.25, meaning that model 1 explains a small amount of variance (*Table 4*). Maximum variance inflation factor (Max. VIF)=1.23, for the variable *Age*, suggesting no collinearity and that overfitting is not an issue and the dispersion test was not significant (p=0.60), meaning that data were not over or underdispersed.

Over four exposures from the first eating event, we found a significant main effect of age, with juveniles 5.39 times (539%) more likely to extract and eat peanuts than adults, and 6.77 times (677%) more likely than infants, but no significant difference between infants and adults (Model 2, *Table 3*). We also found a significant main effect of rank, whereby higher-ranked individuals were more likely to extract and eat peanuts. Specifically, low-ranked individuals were 93% less likely, per unit of standardized rank, than higher-rank individuals to eat. Again, we found no significant effect of sex (Model 2, *Table 3*). Conditional $R^2$=0.55, meaning that model 2 explains a good amount of variance (*Table 4*). Max. VIF = 1.30, for the variable *Age*, suggesting no collinearity and that overfitting is not an and the dispersion test was not significant (p=0.60), meaning that data were not over or underdispersed.

### Muzzle contact rate across repeated exposure to novel food

We recorded a total of 498 muzzle contacts initiated by 64 different individuals in all four study groups during the first four exposures from the first eating event.

**Table 3.** Models outputs for binomial and Poisson generalized linear mixed models.

| Model no. | Outcome | Predictors* | Coefficient | Odds ratio | SE | z- value | p-value [†] |
|---|---|---|---|---|---|---|---|
| 1 | Eat at first exposure with eating event: yes/no (binomial) N=161 | Age: Infant – Adult [‡] | –0.27 | - | 0.65 | –0.41 | 0.912 |
| | | Juvenile – Adult [‡] | 1.43 | 4.17 | 0.52 | 2.74 | *0.017* |
| | | Juvenile – Infant [‡] | 1.69 | 5.43 | 0.67 | 2.51 | *0.032* |
| | | Sex (M) | 0.57 | - | 0.42 | 1.36 | 0.175 |
| | | Standardized rank | –1.52 | - | 0.81 | –1.89 | 0.058 |
| 2 | Eat over four exposures from first eating event: yes/no (binomial) N=161 | Age: Infant – Adult [‡] | –0.23 | - | 0.55 | –0.42 | 0.908 |
| | | Juvenile – Adult [‡] | 1.68 | 5.39 | 0.55 | 3.09 | *0.006* |
| | | Juvenile – Infant [‡] | 1.91 | 6.77 | 0.63 | 3.01 | *0.007* |
| | | Sex (M) | 0.18 | - | 0.41 | 0.45 | 0.656 |
| | | Standardized rank | –2.69 | 0.07 | 0.81 | –3.34 | *<0.001* |
| 3 | Freq. muzzle contact per individual per exposure (Zero-Inflated Poisson) N=256 | Exposure no. | | - | | | |
| | | No. eating (std.) | –0.75 | - | 0.10 | –7.20 | *<0.001* |
| | | Exposure no. | –0.56 | - | 0.21 | –2.70 | *0.007* |
| | | X no. eating | 0.40 | | 0.10 | 3.92 | *<0.001* |
| 4 | Frequency of muzzle contacts initiated (Poisson) N=253 | Prior knowledge (K) | –0.46 | 0.63 | 0.09 | –5.035 | *<0.001* |
| | | Sex (M) | –0.03 | - | 0.29 | –0.11 | 0.911 |
| | | Standardized rank | –2.54 | 0.08 | 0.57 | –4.42 | *<0.001* |
| | | Age: Infant – Adult [‡] | 0.39 | - | 0.43 | 0.90 | 0.635 |
| | | Juvenile – Adult [‡] | 1.79 | 6.00 | 0.36 | 5.02 | *<0.001* |
| | | Juvenile – Infant [‡] | 1.40 | 4.07 | 0.45 | 3.09 | *0.006* |
| 5 | Frequency targeted by muzzle contacts (Poisson) N=253 | Prior knowledge (K) | 1.14 | 3.13 | 0.10 | 11.67 | *<0.001* |
| | | Sex (M) | 0.88 | - | 0.45 | 1.96 | 0.050 |
| | | Standardized rank | –1.56 | - | 0.85 | –1.84 | 0.065 |
| | | Age: Infant – Adult [‡] | –3.44 | 0.03 | 1.03 | –3.98 | *<0.001* |
| | | Juvenile – Adult [‡] | –0.52 | - | 0.51 | –0.94 | 0.603 |
| | | Juvenile – Infant [‡] | 2.92 | 18.6 | 0.93 | 3.13 | *0.005* |

*Reference categories are Adult, Female, and Naïve for categorical predictors: age, sex, and knowledge, respectively; abbr.: N=naïve; K=knowledgeable; M=male.

[†]Bold italics show significant p-values at 0.05 level.

[‡]Indicates post-hoc multiple comparisons (with Tukey correction).

**Table 4.** Variance and standard deviation of random effects and marginal and conditional R squared of the five generalized linear mixed models presented in the paper.

| | Random effects | Variance | Standard deviation | R² marginal | R² conditional | Sample sizes |
|---|---|---|---|---|---|---|
| **Model 1** *Eat at 1st expo w/eating event* | Group | 0.42 | 0.65 | 0.14 | 0.24 | 161 |
| **Model 2** *Eat all expos* | Group | 3.21 | 1.79 | 0.12 | 0.55 | 161 |
| **Model 3** *MC rate per ind, per expo* | Individual | <0.001 | 0.91 | 0.55 | 0.91 | 256 |
| | Group | <0.001 | <0.001 | | | |
| **Model 4** *Freq. MC initiated per ind* | Individual | 1.71 | 1.31 | 0.12 | 0.95 | 253 |
| | Group | 3.42 | 1.85 | | | |
| **Model 5** *Freq. MC received per ind* | Individual | 3.73 | 1.93 | 0.28 | 0.98 | 253 |
| | Group | 1.58 | 1.26 | | | |

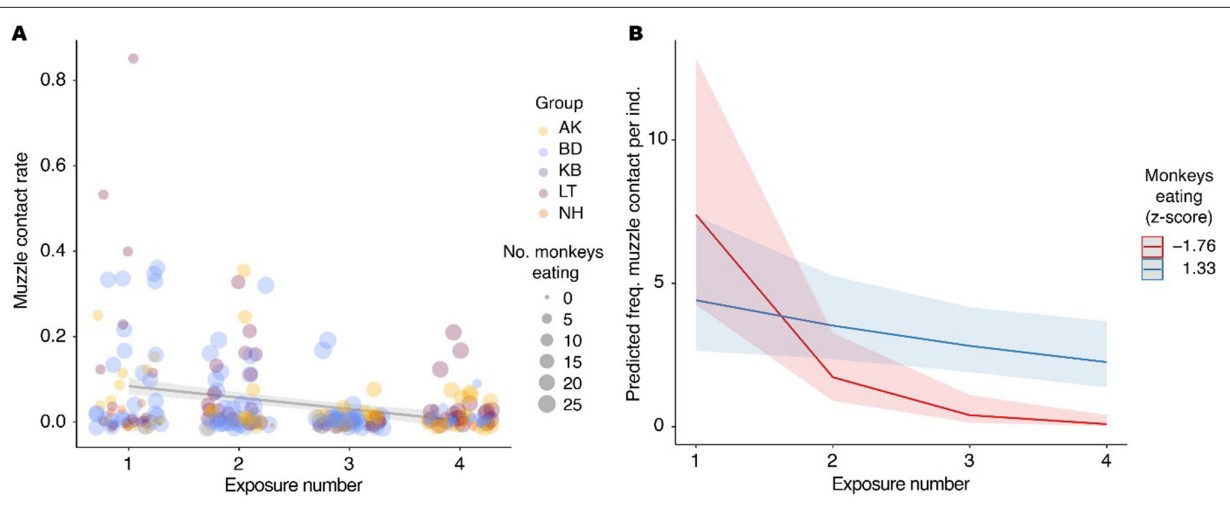

**Figure 3.** Muzzle contact rate across exposures. (**A**) Variation in muzzle contact rate according to the number of monkeys eating and exposure number. Shading shows 95% CI. (**B**) Model predictions based on the significant interaction between exposure number and number of monkeys eating. When greater numbers of monkeys are eating (blue) the effect of exposure number is less extreme than when fewer monkeys are eating (red).

Regarding the rate of muzzle contacts across exposures, we found a significant interaction between the number of monkeys eating and exposure number (Model 3, *Table 3*). The main effects of exposure number and the number of monkeys eating were also significant but are not interpretable in this model due to the presence of the interaction. The significant interaction shows that the effect of exposure number depends on the number of monkeys eating (*Figure 3*). In *Figure 3A and B*, we can see that whilst muzzle contact rate decreased across the exposures, this decrease was less extreme when more monkeys were eating. This suggests that whilst muzzle contacts decreased as the food become, overall, more familiar in the group, it was still dependent on how many opportunities for muzzle contacts there were, i.e., the number of individuals eating, suggesting a potential social function of muzzle contacts. Conditional $R^2$=0.91, meaning that model 3 explains a significant amount of variance (*Table 4*). Max. VIF = 4.36, for the variable *no. monkeys eating (z-score)*, suggesting low collinearity and the dispersion test was not significant (p=0.57), meaning that data were not over or underdispersed.

## Influence of knowledge of novel food and socio-demographic variation on muzzle contact behavior

For all individuals present in groups during the experiment, across the four exposures from the first eating event, the mean (s.d.; range) number of muzzle contacts each naïve individual initiated was 2.30 (4.79; 0–26) and they were targeted 0.91 (3.21; 0–22) times; and each knowledgeable individual initiated 2.20 (4.01; 0–26) muzzle contacts and they were targeted 4.81 (11.18; 0–79) times.

We found significant main effects of prior knowledge, age, and rank on the frequency of initiating muzzle contacts (Model 4, *Table 3*). The number of muzzle contacts initiated by knowledgeable individuals, who had already extracted and eaten peanuts, was reduced by 37% compared to the number initiated by naïve individuals. The odds of lower-ranked individuals initiating muzzle contacts were 92% lower than higher-ranked individuals, per unit of standardized rank. Post-hoc multiple comparisons between age categories showed that juveniles initiated six times (600%) more than adults did and 4.07 times (407%) more than infants did. Conditional $R^2$=0.95, meaning that model 4 explains a significant amount of variance (*Table 4*). Max. VIF = 1.26, for the variable *age*, suggesting no collinearity and no overfitting and the dispersion test was not significant (p=0.63), meaning that data were not over or underdispersed.

Regarding being targets of muzzle contacts, we found significant main effects of knowledge and age, and trends for effects of sex and rank (Model 5, *Table 3*). Here, knowledgeable individuals who had succeeded to extract and eat peanuts were targeted 3.13 times (313%) more than naïve individuals were; juveniles were targeted 37.9 times (3790%) more than infants were, the odds of infants

being targeted were 98% less than adults, and there was no significant difference between juveniles and adults (Model 5, *Table 3*). Conditional $R^2$=0.98, meaning that model 5 explains a significant amount of variance (*Table 4*). Max. VIF = 1.18, for the variable *age*, suggesting no collinearity and no overfitting, and the dispersion test was not significant (p=0.98), meaning that data were not over or underdispersed.

## Discussion

By exposing five groups of wild vervet monkeys to a novel extractive foraging problem, we created conditions under which to (1) observe innovation by individuals, and the uptake of transmission of knowledge within and between groups, and (2) assess the function and patterns of muzzle contact behavior in the context of encountering a novel food. We found evidence of immigrant males as fast innovators, and as potential vectors of information between groups. We observed faster uptake of the innovation in groups when new immigrant males, rather than infants or juveniles, ate first. We found effects of age and rank on uptake of the food, both during the first exposure, and over four exposures, with juveniles and high-rankers eating the novel food more readily than adults and low-rankers. Furthermore, as groups had more exposure to the food, if many monkeys had started to eat peanuts and olfactory contact with novel food increased, the rate of muzzle contacts decreased. Initiating muzzle contacts was influenced by prior knowledge of the food, age, and rank, and being targeted by muzzle contacts was influenced by knowledge and age. Below we discuss the contributions of these results to perspectives on the potential value of dispersing individuals in the innovation and transmission of behavioural adaptations to novel circumstances within populations.

### Who innovated and how did it affect the extent to which the innovation was adopted by the group?

In the two groups where innovation occurred at the first exposure to peanuts (LT and NH), immigrant males, each with less than three months tenure in the group, were the innovators. In KB, an infant innovated, but only at their group's third exposure to the novel food. Fast innovation (i.e. at the first exposure) to exploit a novel resource by new immigrants could be linked to a physiological state related to dispersal but we remain cautious here as our N=2. In the first exposure in $AK_{19}$ there was a relatively new male, *Boc* (*Supplementary file 1*), who had immigrated within four months, but he was very old (>12 years old) and had recently become very inactive. *Boc* disappeared, presumably due to natural death, two months after $AK_{19}$'s exposure to peanuts. Such characteristics may counteract any effects of recent dispersal on exploratory tendencies. Indeed, very old age has been found to be related to declining boldness – a personality trait related to exploration – in big horn ewes (*Réale et al., 2000*). In addition, the group did not show sustained interest in the peanuts, and after brief inspections of the contents of the box, started traveling away from the experiment area within 5 min from the start of the experiment.

Dispersal has previously been associated with exploration and boldness (i.e. low neophobia) in several taxa, with associated neurochemical variation, both within (ontogenetically) and between individuals (*Cote et al., 2010*). Moreover, evidence links lower serotonergic activity with earlier dispersal in rhesus macaques (*Kaplan et al., 1995*), greater social impulsivity in vervet monkeys (*Fairbanks, 2001*), and reduced harm avoidance in humans (*Hansenne and Ansseau, 1999*); all of which together relate to low neophobia, or novelty-seeking, with dispersal. Evidence does not, however, suggest that the dispersing sex in wild vervet monkeys are more bold or explorative overall (*Blaszczyk, 2017*), and in the present study long-term resident males did not show increased interest in the novel food. In another population, long-term resident males also showed reduced responses to novel foods compared to other age-sex classes (*Nord, 2021*). We suggest rather that the unique individual, social, and environmental factors that prompt a male to disperse (*L'Allier, 2020*) may trigger a transitory exploratory behavioral syndrome (*Sih et al., 2012*) that may subside again once males acquire more secure residency in a group. Since dispersal inherently involves heightened risk, periods of long-term residency would be well-served by a state characterized by reduced exploration and increased neophobia to balance the costs of risk-taking over the lifetime. The large variation in risky predator inspection by adult male vervet monkeys, compared to adult females found in *Blaszczyk, 2017* also supports this. In the case of *Boc*, being likely near the end of his lifetime, this could also explain why

he did not innovate, however further data are needed to understand whether this is a valid explanation or not. Future work focusing on the behavior of dispersing individuals at multiple time points, both proximal and distal to dispersal events, in this species, and others, will help to more conclusively address this hypothesis. We highlight the need for researchers to consider the nuances of life-history characteristics beyond simply splitting by broad age-sex categories.

We found that when immigrant males were first to extract and eat a novel food in their new groups, during the first exposure to it in BD, LT, and NH, and during the second exposure in $AK_{20}$, other monkeys quickly followed them in doing so. As discussed above, two of these cases (LT, NH) involved innovation by the immigrant males, and in BD and $AK_{20}$, the males had learned to extract and eat the food in their previous groups. In contrast, following innovation by the infant in KB at their third exposure, no other individuals followed in extracting and eating peanuts during that exposure. Over the three subsequent exposures that followed these initial eating events, very few monkeys started to extract and eat peanuts in KB, whereas in BD, LT, and AK, where new immigrants ate first, large proportions of these groups learned to extract and eat peanuts. In NH however, this was not the case and is of great interest because though a new immigrant male innovated at the first exposure, he left the group before the second exposure. Closer inspection of our data revealed that only juveniles had started extracting and eating peanuts after the male in the first exposure and were thus the only knowledgeable individuals at the second exposure. Similarly, in KB, only juveniles ate after the infant innovated. Our interpretation of these results is that immigrant males were more effective in facilitating group members to overcome neophobia towards a novel food than infants or juveniles, which is in line with studies reporting age-biased social learning (*Barrett et al., 2017*; *Canteloup et al., 2021*). Nonetheless, in NH, more individuals did eventually start to eat during their fourth exposure, including a high-ranked adult female (after which the innovation spread rapidly, resulting in the data presented in *Canteloup et al., 2021*). In NH, the juveniles eating at the beginning were older than the infants and 1-year-old who started eating in KB. It is possible that age bias is less strong for older juveniles since they should have more reliable knowledge than their very young counterparts. Alternatively, this difference between NH and KB could be because the juveniles eating earlier in NH were of high rank, whereas the infants and juveniles who began to eat in KB were of low rank. Indeed, rank-biased social learning has also been found in previous work in two groups (KB and NH) of this study population (*Canteloup et al., 2021*; *Canteloup et al., 2020*). It is nonetheless likely that various interactions of socioecological factors affect the influence of juveniles in overcoming food neophobia in the wild, and it still took repeated exposures before groupmates of the NH juveniles began eating.

Alternative explanations for these patterns of uptake, such as group habituation level to humans or the different experimental histories of each group (see STRANGE framework: *Webster and Rutz, 2020*), are unlikely. Indeed, in two groups with extensive experimental history (NH, AK), few individuals ate the novel food at their first exposure, whilst in the least habituated group (LT), with the most minimal experimental history (*Forss et al., 2022*), a great proportion of individuals adopted the novel food at the first exposure (*Figure 2A*). We hypothesize that observing new immigrants eat the novel food triggered groupmates to try it, rather than factors related to group idiosyncrasies that would be expected according to the STRANGE framework (*Webster and Rutz, 2020*).

Moreover, whilst previous experiments suggested that high-ranked adult philopatric females are preferred over high-ranked adult males as models to learn from *van de Waal et al., 2010*, in the context of exploiting a novel resource, risk dynamics come into play. Adult females are likely to be the most risk-averse age-sex category, due to the great potential negative impact of risks on their inclusive fitness, especially when young dependent offspring are present or whilst pregnant. This might limit their potential to discover new information that others can exploit. Under these conditions, adult males that are either in an exploratory dispersal state, or that enter a group with knowledge of resources previously unknown to the group (as in *McDougall et al., 2010*) may play important roles in generating and/or facilitating the spread of behavioral adaptations to exploit novel resources and face rapid environmental changes.

## Socio-demographic variation in the uptake of the novel food

Both, in the first exposure, and over four exposures, juveniles were more likely to eat than adults and infants. These results suggest that juveniles overcome neophobia faster, corresponding closely with results regarding risk-taking in another population of vervet monkeys (*Fairbanks, 1993*). Furthermore,

juvenile vervets have been found to learn faster (*Kumpan et al., 2020*), and work on other species suggests that juveniles are overall more exploratory and less neophobic than adults (*Benson-Amram et al., 2013*; *Bergman and Kitchen, 2009*; *Miller et al., 2015*; *Sherratt and Morand-Ferron, 2018*). Taken together we propose that juveniles are, in general, more prone to taking risks around novelty, particularly when conspecifics provide social information. Moreover, alongside the results of section 1 a, we propose that it could be adaptive that groups do not follow novel foraging information from juveniles as readily as adults (i.e. in NH), as this may limit the spread of potentially dangerous information acquired by exploratory but inexperienced juveniles. We also expect that infants were not more likely than adults to eat due to still being at least partly reliant on their mothers to learn their foraging repertoire (*Whiten and van de Waal, 2018*) in contrast to juveniles who explore more independently.

Over four exposures, higher-ranked individuals were significantly more likely to eat than lower-ranked individuals, probably due to preferential access to the resource as it became more familiar and sought-after.

## Muzzle contact frequency in groups

Muzzle contact rates decreased over repeated exposure to the novel food, providing some support for our hypothesis that the less muzzle contacts would occur when the food had become more familiar in each group. However, we expected this effect to be greatest when many monkeys were eating. Contrary to this, muzzle contact rates decreased more slowly when more monkeys were eating. This also makes sense, because more monkeys eating means more monkeys were in the area of the novel food, and therefore, there were more opportunities to engage in muzzle contact. The steeper decrease in muzzle contact rate when fewer monkeys were eating also likely reflects that there were more muzzle contacts at the very beginning, when very few monkeys were eating, and the later exposures where very few monkeys were eating also gave rise to fewer opportunities for muzzle contacts. Cases where very few monkeys were eating in later exposures were due to KB and NH, where very few individuals started to eat over the four exposures time frame examined here, and the two exposures in BD (Expo. 4) and LT (Expo. 3) with only small portions of the group present. Nonetheless, the overall decrease in muzzle contact rate demonstrates the relevance of the behavior in the context of an unknown foraging item, because as the monkeys became more familiar with it by eating it, they sought olfactory information from their conspecifics less frequently. This result concurs with findings from a similar study in wild olive baboons (*Laidre, 2009*). It could be argued that our conclusion regarding muzzle contact serving to acquire information is premature in the absence of evidence that muzzle contact directly led to individuals eating. However, unlike in the context of observing and learning to use novel tools (e.g. *Hobaiter et al., 2014*), we do not expect muzzle contact to be a prerequisite to learning to extract and eat peanuts. We argue that muzzle contacts need not be correlated with extracting peanuts in such a manner in order to support that they serve to acquire information. We provide further evidence to support this function below (section 2b).

That there were more muzzle contacts when more monkeys were eating could be interpreted that muzzle contacts are provoked by seeing conspecifics consume any food, regardless of its novelty. We have, however, used provisions of corn kernels in experiments for 10 years with this study population, and when presenting monkeys with this now familiar resource, we do not see rates of muzzle contact anywhere close to those observed during the early exposures in this experiment (*Rochat, 2022*). This is supported by the significant main effect of exposure number (*Figure 3*).

## Influence of knowledge of novel food and socio-demographic variation on muzzle contact behavior

Muzzle contacts were initiated the most by individuals that had not yet extracted and eaten peanuts (hereafter, naïve individuals; opposite: knowledgeable), higher-ranked individuals, and juveniles. Contrastingly, muzzle contacts were targeted the most towards knowledgeable individuals, and the least towards infants. We find the most compelling evidence for our hypothesis of the function of muzzle contact in information acquisition in that naïve individuals initiated the most and knowledgeable individuals were targeted the most. We do not make claims related to knowing what others know, but rather we assume that seeing a group member eating an unknown resource prompts the initiation of muzzle contact toward that individual. Moreover, this result corroborates the finding in (2a) of

decreasing muzzle contact frequency with increased exposure to and familiarity with the resource, and the overall function of muzzle contact in soliciting foraging information.

The effect of age on initiating muzzle contacts falls in line with the expected direction of information transfer from older to younger individuals (*Whiten and van de Waal, 2018*), with juveniles initiating the most (as also found in *Drapier et al., 2002*; *Chauvin and Thierry, 2005*; *Nord et al., 2021*). It also corroborates general findings regarding juveniles' novelty seeking and faster learning (e.g. *Kumpan et al., 2020*; *Benson-Amram et al., 2013*; *Bergman and Kitchen, 2009*; *Miller et al., 2015*; *Sherratt and Morand-Ferron, 2018*) as discussed above. However, that adults were not targeted significantly more than juveniles in this study (as in *Drapier et al., 2002*; *Chauvin and Thierry, 2005*; *Nord et al., 2021*) is probably because juveniles were more likely to become knowledgeable of the novel food in this experiment (section 1b), and were, therefore, targeted more. This may seem contradictory to our assertion above, that individuals would adaptively not follow information from juveniles, however it is also possible that there is a critical mass effect, whereby when many individuals are already consuming a novel resource, juveniles may become valid sources of information. This is, however, beyond the scope of the present study, but requires further investigation. Furthermore, that infants were targeted the least does follow the direction of information transfer from older to younger individuals, and complements our finding that when an infant innovated, the innovation was not taken up widely in the group.

That high-ranked individuals were more likely to be both initiators and targets is likely because, first, like juveniles, they were far more likely to become knowledgeable, and second, because a high degree of tolerance is required by the target towards the initiator due to the close proximity in which this behavior occurs (as described in *Nord et al., 2021*). Lower-ranked individuals are not tolerated at the close proximity required to initiate muzzle contacts, especially around food resources; and they were much less likely to become knowledgeable, likely reducing their salience as targets.

## Conclusion

We add to the literature an experimental example of exploitation of a novel resource by multiple groups, facilitated here by dispersers. Our results provide evidence that dispersing individuals may promote the generation of new, environmentally relevant information and its spread around populations – a factor that has been largely overlooked, despite the known role of dispersal in gene flow (*Greenwood, 1980*). We urge future research to investigate what physiological mechanisms might exist underpinning a transitory dispersal syndrome characterized by heightened exploration and reduced neophobia that is triggered during, or triggers, dispersal. We studied a species with sex-biased dispersal and we open up the question of whether similar dynamics as suggested here might be at play in species where both sexes disperse, and whether dispersing females and males show similar levels of boldness during dispersal or not, due to different life-time risk mitigation strategies. Finally, we suggest further research, in diverse species, into whether dispersers transmit valuable information between groups, which can have major implications for population fitness, especially in the context of the rapid anthropogenic change that most animal populations now face. This study contributes novel insights into the roles of dispersers in wider behavioral ecology, which we hope will inspire and inform future work, spanning the disciplines of animal behavior and cultural evolution.

## Materials and methods
### Experimental model and subject details

The study was conducted at the 'Inkawu Vervet Project' (IVP) in a 12000-hectares private game reserve: Mawana (28°00.327 S, 031°12.348E) in KwaZulu Natal province, South Africa. The biome of the study site is described in *Bono et al., 2018*.

Five groups of habituated wild vervet monkeys (*Chlorocebus pygerythrus*) took part in the study: 'Ankhase (AK)', 'Baie Dankie (BD)', 'Kubu (KB)', 'Lemon Tree (LT)', and 'Noha (NH)'. Habituation began in 2010 in AK, BD, LT and NH, and in 2013 in KB. All observers in the field were trained to identify individuals by individual bodily and facial features (eye rings, scars, color, shape etc.). During the study period, these stable groups comprised between 19 and 65 individuals including infants (*Table 1*). We refer to the group AK differentially as $AK_{19}$ and $AK_{20}$, representing their status in 2019 and 2020,

respectively, as 40% of the group composition changed between years due to dispersals, deaths, and changes in age categories (infants that became juveniles; see *Supplementary file 1a*).

## Dominance rank calculations

Agonistic interactions (aggressor behavior: stare, chase, attack, hit, bite, take place; victim behavior: retreat, flee, leave, avoid, jump aside) were collected ad libitum (*Altmann, 1974*) on all adults and juveniles of each group. These data were collected for a duration of one year, up until the date of each group's first exposure, during all behavioral observation hours and during experiments involving food provisions. Data were collected by CC, PD, and different trained observers from the IVP team. Before beginning data collection, observers had to pass an inter-observer reliability test with Cohen's kappa >0.80 *McHugh, 2012* for each data category between two observers. Data were collected on handheld computers (Palm Zire 22) using Pendragon software version 5.1 and, from the end of August 2017, on tablets (Vodacom Smart *Table 2*) and smartphones (Runbo F1) equipped with the Pendragon version 8.

Individual ranks were calculated using the I&SI method (*de VRIES, 1998*), based on win/lose outcomes of dyadic agonistic interactions, using Socprog software version 2.7. Linearity of hierarchies are reported in *Supplementary file 1b*. Ranks were standardized to represent the proportion of the group that outranks each individual, falling between 0 (highest) and 1 (lowest) in each group (rank – 1/ group size). Agonistic data on adults and juveniles were included, and we assigned infants with the rank just below their mother, based on the youngest offspring ascendency in this species (*Cheney and Seyfarth, 1990*).

## Peanut exposures

We provided each group with a highly nutritious novel food that required extraction before consumption – unshelled peanuts (*Figure 1A*) – in large quantities to avoid monopolization by single individuals. Experiments took place after sunrise when the monkeys were located at their sleeping site during the dry, food-scarce South African winter, to maximize both the motivation to engage in food-rewarded experiments and the number of group members in the vicinity.

CC ran field experiments from May-June 2018 in KB and NH, and PD led the experiments during August-September 2019 in AK ($AK_{19}$), BD and LT, and May-June 2020 in AK ($AK_{20}$; *Supplementary file 1*). $AK_{19}$ and $AK_{20}$ are used to denote AK in 2019 and 2020, respectively, because approximately 40% of the group composition changed in the year in between exposures, and no individuals ate in 2019, meaning the food was still novel to the group in 2020. *Figure 1B* illustrates the relevant male immigrations into and emigrations out of these groups. *Avo* left his natal group KB to immigrate to NH two weeks before their first experiment in 2018. *Avo* never ate peanuts before the first exposure in NH. KB had no new males since 2017. *Pro* originated from NH and learned to eat peanuts during their experiment in 2018. He immigrated to BD three weeks before their first experiment in 2019. *Bab* immigrated into LT six weeks before their first experiment in 2019, from an unhabituated group, though he was habituated to humans and food experiments due to previous residence in two habituated study groups. *Bab* never ate peanuts before LT's first exposure. In 2020, two males, *Twe* and *Yan*, who were present in NH during peanut exposures in 2018, immigrated to AK, six and ten weeks, respectively, before their experiment in 2020. *Twe* ate peanuts in NH during peanut exposures that continued beyond the four presented here in a previous study (*Canteloup et al., 2021*), and *Yan* had observed many others eating peanuts. *Yan* was the first to eat in AK in 2020 (*Figure 1B*).

Peanuts were presented to all groups in clear rectangular plastic boxes (34 × 14 × 12 cm), containing 1–2.5 kg of unshelled peanuts. We considered the beginning of an exposure when the experimenters placed the box on the ground, removed the lid, and stepped away, giving access to the monkeys. Exposures ended when the monkeys were clearly traveling away from the experiment site. Experimental sites were chosen at whichever sleeping site the monkeys were found, except in BD, in which it was wherever the group was after 1 hr of a focal follow of *Pro*, due to previous aims of the study, and thus was not always at the sleeping site. The boxes were placed visible to as many group members as possible, with the exception of the first exposure in BD where we placed the box close to the knowledgeable male, *Pro*, due to our initial aim to investigate intergroup transmission. One box of peanuts was offered per exposure in AK, BD, and LT. In KB and NH, two boxes were offered during each exposure, and were topped up when they were empty. Exposure durations ranged from 5 min ($AK_{19}$)

to 74 min (LT). KB and NH had 10 exposures on 10 different days; $AK_{20}$, BD, and LT had four exposures on four different days; and $AK_{19}$ had a single exposure (*Supplementary file 1*). The groups tested by PD (AK, BD, LT) had fewer exposures overall due to time constraints. Here, we present results for each group from the first four exposures *from the first eating event* in each group. Whilst AK, NH, and KB had more than four exposures in total, BD and LT had only four, meaning that taking four exposures from the first eating event is the most reasonable way to compare these groups. In addition, after four exposures from the first eating event, over 90% of AK and LT had learned to eat peanuts, limiting the reasons to run further exposures with them.

Reactions to and interactions with peanuts were recorded by three to five observers using handheld JVC video cameras (EverioR Quad Proof GZ-R430BE) and cameras mounted on a tripod. Observers narrated the identities of monkeys interacting with peanuts for later video coding.

## Quantification and statistical analysis

### Video coding

To extract the identities of individuals who successfully extracted and ate peanuts from their shells during each exposure, PD-coded videos of $AK_{19-20}$, BD, and LT with the Windows 10 default video software, and CC-coded videos of KB and NH with Media Player Classic Home Cinema software version 1.7.11. Having extensive experience working in the field with these groups, PD and CC were proficient in recognizing individuals from the videos, and often the identities were narrated live in the audio of the video recordings which provided additional assurance of accuracy.

For analyses of muzzle contacts, GL and PD counted the frequency of muzzle contacts in videos of $AK_{20}$, BD, KB, LT, and NH. MC assigned identities of individuals involved in these muzzle contacts using data provided by PD in the form of scan samples of the identities of all monkeys on the screen from left to right at every minute of each video.

To test interobserver reliability, PD recorded 15% of all videos in the study that were originally coded by CC, to verify agreement on what each coded as 'successful extracting and eating,' and achieved a Cohen's kappa of 0.96. PD also recorded 10% of the videos of the study that were originally coded by GL to verify agreement on what constituted muzzle contact interactions, and achieved a Cohen's kappa of 0.98.

**Table 5.** Model structures.

| | Distribution | Outcome | Fixed effects | Random effects |
|---|---|---|---|---|
| **Model 1**<br>*Eat at 1st expo w/ eating event* | Binomial | Eat: Yes / No | Age (adult/juv./infant)<br>Sex (F/M)<br>Rank* | Group |
| **Model 2**<br>*Eat all expos* | Binomial | Eat: Yes / No | Age (adult/juv./infant)<br>Sex (F/M)<br>Rank* | Group |
| **Model 3**<br>*MC rate per ind, per expo* | Zero-inflated Poisson | Freq. initiated | Exposure number (1-4)<br>No. monkeys eating (z-score)<br>Duration of exposure (mins.; offset) | Group<br>Individual |
| **Model 4**<br>*Freq. MC initiated per ind* | Poisson | Freq. initiated | Prior success (1/0)<br>Age (adult/juv./infant)<br>Sex (F/M)<br>Rank*<br>Total exposure duration per ind. (mins.; offset) | Group<br>Individual |
| **Model 5**<br>*Freq. MC received per ind* | Poisson | Freq. received | Prior success (1/0)<br>Age (adult/juv./infant)<br>Sex (F/M)<br>Rank*<br>Total exposure duration per ind. (mins.; offset) | Group<br>Individual |

*Dominance rank calculated with I&SI method, and standardized between 0 (high rank) –1 (low rank) – see Methods for more details.

## Data analysis

### Who innovated and how did it affect the extent to which the innovation was adopted by the group?

We did not formally analyze these data, we only described who innovated and which individuals began to consume the novel food in each group.

### Demographic variation in the uptake of the novel food

We used two generalized linear mixed models (GLMMs) for data following a binomial distribution to investigate demographic variation in whether or not individuals extracted and ate peanuts (question 2b). The first model investigated (i) the first exposure with an eating event (AK$_{20}$, BD, KB, LT & NH; Model 1; *Table 5*), and the second model investigated (ii) four exposures from the first eating event (AK$_{-20}$, BD, KB, LT & NH; Model 2; *Table 5*). In each model, the outcome was a binomial yes/no variable (did the individual eat), we considered age, sex, and rank (standardized rank) as fixed effects and the group was included as a random effect. Males that dispersed between groups were only considered in their first group in this analysis, so all individuals were only considered once in these models. Effect sizes are reported as odds ratios. Here and in all subsequent models, we inspected Q-Q plots and residual deviation plots, and tested for over-/underdispersion using the DHARMa R package (*Hartig, 2022*) to assess model suitability.

### Rate of muzzle contact over repeated exposure to the novel food

To investigate the effect of exposure to the novel food on muzzle contact rate, we looked at four exposures from the first eating event for each group, as this event marks when at least one member of the group had recognized the novel food as a viable food. We also wanted to account for the number of individuals eating during each exposure, as it was inherent in our hypothesis that muzzle contacts around the novel food would be related to individuals eating it. Specifically, we expected the muzzle contact rate to decrease across exposures when there were many monkeys eating it and, therefore, developing their own knowledge of it, but not if only very few were eating. To test this, we fitted a Zero-Inflated Poisson GLMM (using the glmmTMB function from R package 'glmmTMB' *Brooks et al., 2017*) with a frequency of muzzle contacts initiated by each individual as the outcome variable, exposure number, and the number of monkeys eating during the exposure (z-transformed) as fixed effects, with an interaction between the two, and group and individual as random effects. We included the log of the duration of the experiment as an offset in order to model the rate of muzzle contacts per minute per individual (Model 3; *Table 5*). Effect sizes are reported as odds ratios. We used the DHARMa R package (*Hartig, 2022*) to assess model suitability (as above), and to test zero inflation in an initial Poisson GLMM.

### Influence of knowledge of novel food and socio-demographic variation in muzzle contact behavior

We wanted to assess which factors influenced individuals' involvement in muzzle contact interactions. Specifically, we wanted to test hypotheses regarding the function of this behavior in information acquisition, so whether individuals' prior knowledge of the food was an important factor or not. We expected individuals who had not yet successfully extracted and eaten peanuts to initiate more muzzle contacts, and those who had already successfully extracted and eaten peanuts to be targeted more. In addition, if muzzle contact is involved in information acquisition, as we predicted, we would also expect variation between different age, sex, and rank classes in whether they initiated more or were targeted more in line with our current state of understanding of social learning in this species. To investigate this, we counted how many muzzle contacts each individual of each group was involved in, first separated by whether they were the initiator or target, and further, by whether they were naïve or knowledgeable about the novel food. We then used two GLMMs to analyze (i) what factors influenced *initiating* muzzle contacts (Model 4), and (ii) what factors influenced *being targeted* by muzzle contacts (Model 5). Model 4 had the frequency of initiating as the outcome variable, with prior knowledge, age, sex, and standardized rank as predictors, and individual and group as random effects (*Table 5*). Model 5 had the frequency of being targeted as the outcome variable, with prior knowledge, age, sex, and standardized rank as predictors, and individual and group as random effects (*Table 5*). In both

of these models, the log of each group's total duration (minutes) of exposure to peanuts was used as the offset, and effect sizes for both of these models were assessed as odds ratios.

We ran post-hoc multiple comparisons (with Tukey correction) between the age categories (adult/juvenile/infant) using estimated marginal means comparisons from the 'emmeans' R package (*Lenth, 2021*).

In all analyses described above, we probed interactions using post-hoc multiple comparisons (with Tukey correction) of estimated marginal means using the R package 'emmeans' (*Lenth, 2021*) and plotted the interactions using the R package 'interactions' (*Long, 2019*). All model diagnostics were analyzed using the 'DHARMa' R package (*Hartig, 2022*), and multicollinearity was assessed using variance inflation factors (VIFs) calculated with package 'car.' We calculated marginal and conditional r-squared for each model using the MuMIn package in R and compared their value to assess the amount of variance explained by the fixed effects only ($R^2$ marginal) and by the fixed effects and the random effects ($R^2$ conditional). We also compared the AICs of the models with and without random effects and the lowest AICs were always those of the models with random effects, meaning that the models that fitted better to our data were those with random effects. Model assumptions were satisfied unless otherwise reported and adjustments made. Statistics were computed in R Studio (R version 4.0.3), and linear regression was done with the base R stats package (*R Development Core Team, 2020*). GLMMs were done using the 'lmerTest' package (*Kuznetsova et al., 2017*).

All data and R scripts are made available at: https://doi.org/10.5281/zenodo.7376673.

## Acknowledgements

This study was supported by the Swiss National Science Foundation (PP03P3_170624 and PP00P3 198913), the Branco Weiss Fellowship – Society in Science, granted to Erica van de Waal, and by the Fyssen Foundation and the Fondation des Treilles granted to Charlotte Canteloup. At the time of revisions, Erica van de Waal was supported by the European Research Council under the European Union's Horizon 2020 research and innovation program for the ERC 'KNOWLEDGE MOVES' starting grant (grant agreement No. 949379) and Charlotte Canteloup was supported by the CNRS. We are grateful to the van der Walt family for their permission to conduct the study on their land and to Arend van Blerk and Michael Henshall for their support in the field. We are particularly thankful to Mabia Biff Cera, Adam Cogan, Adwait Deshpande, Sashimi Wieprecht, Manon Kerréveur-Lavaud, Varun Manavazhi, Maria Teresa Martinez Navarrete, Tecla Mohr, Aurora Rozmaryn, Claudia Seminara, and Luca Silvestri for their help in data collection. We are grateful to Frédéric Schütz, Rachel Harrison, and Cédric Girard-Buttoz for their advice on statistical analyses. We thank Redouan Bshary, Sofia Forss, Rachel Harrison, Carel van Schaik, and Andrew Whiten for comments on an earlier version of the manuscript.

## Additional information

### Funding

| Funder | Grant reference number | Author |
| --- | --- | --- |
| Schweizerischer Nationalfonds zur Förderung der Wissenschaftlichen Forschung | PP03P3_170624 | Erica van de Waal |
| Schweizerischer Nationalfonds zur Förderung der Wissenschaftlichen Forschung | PP00P3_198913 | Erica van de Waal |
| Branco Weiss Fellowship – Society in Science | | Erica van de Waal |
| Fondation Fyssen | | Charlotte Canteloup |

| Funder | Grant reference number | Author |
|---|---|---|
| Fondation des Treilles | | Charlotte Canteloup |
| Horizon 2020 | 949379 | Erica van de Waal |
| Centre National de la Recherche Scientifique | | Charlotte Canteloup |

The funders had no role in study design, data collection and interpretation, or the decision to submit the work for publication.

## Author contributions

Pooja Dongre, Conceptualization, Data curation, Formal analysis, Investigation, Visualization, Methodology, Writing – original draft, Writing – review and editing; Gaëlle Lanté, Data curation, Formal analysis; Mathieu Cantat, Data curation; Charlotte Canteloup, Conceptualization, Data curation, Formal analysis, Funding acquisition, Investigation, Methodology, Writing – review and editing, Supervision, Writing – original draft; Erica van de Waal, Conceptualization, Resources, Supervision, Funding acquisition, Methodology, Writing – review and editing

## Author ORCIDs

Pooja Dongre http://orcid.org/0000-0001-6957-3972
Charlotte Canteloup https://orcid.org/0000-0001-5462-081X
Erica van de Waal https://orcid.org/0000-0001-7778-418X

## Ethics

Our study was approved by the relevant local wildlife authority, Ezemvelo KZN Wildlife, South Africa (though no reference number was provided by them). The University of Lausanne, Switzerland, did not have an ethics committee for the study of animals in other countries, however, we ensured our research adhered to the "Guidelines for the use of animals in research" of the Association for the Study of Animal Behaviour (available here: doi:10.1016/j.anbehav.2019.11.002).

## Decision letter and Author response

Decision letter https://doi.org/10.7554/eLife.76486.sa1
Author response https://doi.org/10.7554/eLife.76486.sa2

## Additional files

### Supplementary files
- Transparent reporting form
- Supplementary file 1. Supplementary tables 1a and b.

### Data availability

All data and code used for analyses are freely available at: https://doi.org/10.5281/zenodo.7376673.

The following dataset was generated:

| Author(s) | Year | Dataset title | Dataset URL | Database and Identifier |
|---|---|---|---|---|
| Dongre P, Lanté G, Cantat M, Canteloup C, van de Waal E | 2022 | Role of immigrant males and muzzle contacts in the uptake of a novel food by wild vervet monkeys | https://doi.org/10.5281/zenodo.6827878 | Zenodo, 10.5281/zenodo.6827878 |

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
