## [Editor Report]

This important study provides new insights into behavioural mechanisms involved in the transmission of information surrounding innovation in a social species. Combining experimental and observational evidence, the results are solid and convincing regarding the effects of age, rank and muzzle contacts in transmitting knowledge among vervet monkeys. The work will be of interest to ethologists, behavioural ecologists and comparative psychologists.

---

## [Decision Letter]

**Decision letter after peer review:**

Thank you for submitting your article "Role of immigrants and muzzle contacts in the uptake of a novel food by wild vervet monkeys" for consideration by *eLife*. Your article has been reviewed by 3 peer reviewers, and the evaluation has been overseen by a Reviewing Editor and George Perry as the Senior Editor. The following individuals involved in review of your submission have agreed to reveal their identity: Julie Teichroeb (Reviewer #3).

Essential revisions:

All three reviewers found this study to hold great potential in providing significant new insights into the field of social learning and transmission in wild animals. However, due to a number of important concerns with the analysis it is difficult to ascertain whether the authors' claims are valid given the evidence. We invite the authors to address these major concerns in a substantially revised version of the manuscript.

1) Statistical analysis needs revising:

A number of concerns regarding multiple testing and the structure of your mixed models require attention. In particular, please consider using a multimodel approach due to the exploratory nature of your analyses (see suggestions from Reviewer 1 and 2) and revise the structure of your mixed models to include essential random effects where necessary, and address potential confounding variables such as group size, combining age and sex into one variable, and the directionality of muzzle-muzzle contact initiations (see all 3 Reviewer comments for details). Please also ensure the code for all your models/analyses have been provided.

2) Methods need to be more transparent:

All reviewers found that parts of your study lacked sufficient detail to be repeated by others. Could you please provide clear criteria for the various decisions made throughout the study as well as justifications for cut offs used (e.g., why 3 months for immigrant males?). The rank calculation also needs more clarity and various decisions made by the authors are not justified/clear. Please also provide interobserver reliability tests regarding coding (see Reviewer 1 and 2 for details).

3) Study needs reframing:

All three reviewers found the introduction lacked direction and conceptual clarity. Please provide a more thorough rationale for your study and integrate this into a list of explicit research questions and predictions. The discussion would also benefit from consideration of alternative explanations (see details in comments from all 3 Reviewers).

*Reviewer #1 (Recommendations for the authors):*

I have some suggestions regarding the methods:

– It needs to be reported what decided when a trial was begun, and ended, as this would help explain the differences in the trial lengths.

– It would be very helpful to report how STRANGE these animals are, especially given this manuscript is about innovation; see:

Webster, M. M., and Rutz, C. (2020). How strange are your study animals. Nature. Nature, 582, 337-340. https://www.nature.com/articles/d41586-020-01751-5?sf235295265=1

Farrar, B. G., and Ostojić, L. (2020). It's not just the animals that are strange. Learn Behav. Learn Behav. https://doi.org/10.3758/s13420-020-00442-5

I have some suggestions regarding the analyses:

– The R package DHARMa is a great resource for model residual diagnostics.

– There are no effect sizes reported for any of the models.

– The code for the first muzzle contact model, looking at rate, wasn't included, and so I was unable to review it. Further, it was unclear as to if only a subset of the available data was used for this model (groups with lots of eaters), and if so, why. Including group as a random effect could help account for any group differences that the authors may have felt relevant to subsetting the data, thus allowing all of the data to be examined.

– Collecting data in the field is quite different from coding from videos, and reporting a reliability measure on the video data would help readers to assess the manuscript's findings.

– It seems to me that all models need both ID and group as grouping variables (random effects). This is because all of the models, as far as I can tell, include some of the same adult males (though on different troops), and all of the analyses are all conducted across all of the groups, requiring group to be a random effect.

– Exposure to the experiments varied widely across groups, and it is unclear if all animals on the same groups were around for the trials (I'm assuming they weren't as some trials were dropped due to low numbers of participants). These discrepancies need to be accounted for in the manuscript and analyses.

– There is no explanation as to why there are three different versions for the provided models looking at muzzle contact (1 model with (non-normalized) rank, 1 with two interactions but without rank, and 1 with three interactions, but also without rank). Why was rank dropped? Why were interactions included? If multiple models are going to be considered, a model comparison value, like AIC, should be included to compare the models (after first explaining the theoretical reasons as to why different versions were considered).

– How was rank normalized, and why was normalized rank considered in some models but not others?

– Figure 2A y-axis is unclear- shouldn't it be closer to 50% for BD given 35 innovated? Or is this just after the first exposure? Or first eating event?

Other questions/ suggestions:

– Line101: why the possessive on "groups"?

– Line 286: this is in line with the hypothesis of Nord et al. 2020, which concluded that, "both kin and low-rank- ing animals serve as discriminative stimuli for social tolerance and that foraging animals serve as discriminative stimuli for food availability" Though, this manuscripts first muzzle contact model (for which the code was not provided) found no evidence of a rank effect (though rank wasn't included in the interaction models), which is in contrast to Nord et al., thus providing an interesting extension to findings about muzzle contacts, in that social tolerance may play less of a role in cases of novel foods.

– Line 289: This also agrees with Nord et al.

– Line 300-302: Do the juveniles referred to here out rank adults? Are ranks calculated across all age types in these data?

– Line 303-305: The manuscript reports that juveniles have information, but aren't passing it…how is this an adaptation to risky behavior, if juveniles are more likely to eat something novel anyway? Isn't this doubly bad for juveniles, in that they are more likely to eat something they don't know, which can be dangerous, and also aren't functioning as a source of information when they have it?

– 309-311: This could possible be tested putting age and sex, as a combined variable, in the model, creating no need to have an interaction between age and sex and knowledge, and instead would only need one between Age-Sex and knowledge

[Editors' note: further revisions were suggested prior to acceptance, as described below.]

Thank you for resubmitting your work entitled "Role of immigrant males and muzzle contacts in the uptake of a novel food by wild vervet monkeys" for further consideration by *eLife*. Your revised article has been evaluated by George Perry (Senior Editor) and a Reviewing Editor.

The manuscript has been greatly improved but there are some remaining issues that need to be addressed. In particular, the responses to reviewer concerns regarding the conceptual framing of the study and the analysis are not sufficiently addressed and we find the justification for not making the suggested changes unsatisfactory.

In your revised submission please ensure you adequately address the major comments provided by Reviewers 1 (points 1-6) and 2 (points 1 and 2).

To summarize, the introduction still requires substantial revisions to provide a more clear conceptual framework and predictions, along with how this aligns with your analyses. The analysis also requires attention as key components of models are still missing, (i.e., random effects) and the transparency of your results will be much improved if sample sizes and R-squared values are reported for each model. See Reviewer comments below for details.

*Reviewer #1 (Recommendations for the authors):*

The authors have done a great job making the introduction more applicable to the research conducted. It is still a bit disorganized, which I comment on explicitly below. Furthermore, the exploration of role of knowledge in information-seeking is an important contribution, as this kind of social/contextual account is not often a topic of study in the social learning and innovation literature.

1. The introduction seems to present the idea that learning what to eat is innovation, and that dispersing is reflective of innovative abilities, though I'm not certain that this is what the authors intend. In the discussion, the explanation as to why dispersing animals may be more likely to innovate when they are dispersing (a "transitory exploratory behavioural syndrome") is much more clear. Indeed, it might work better to introduce this idea in the introduction, and continue along the same lines with regard to juveniles-that given that they need to learn about their environments, they might be better primed to innovate akin to the transitory behavioural syndrome of dispersing animals, rather than viewing ontogeny as a constant state of innovation. However, it is still unclear as to when males are considered immigrants vs. residents. Surely after group integration, they are no longer experiencing a "transitory exploratory behavioural syndrome.". Explicitly outlining this distinction (dispersing vs. resident) would greatly help the manuscript.

2. Additionally, it is very odd that all of the predictions presented in the introduction are exactly what is found, even when they are in contrast to the literature. This is especially difficult given the predictions about rank, and that, as long as I'm reading the methods correctly, the rank results are actually presented backwards, meaning that the authors found the opposite of what they report (i.e., because increasing rank values equals decreasing rank--higher ranking animals are represented by lower values--and the models found a positive effect of rank, then it is that lower ranking animals were more likely to eat the novel foods, not higher ranking animals; this is in line with the neophobia literature that predicts that higher ranking animals should be more neophobic, in contrast with the prediction provided in the introduction).

I have a number of questions with regard to the analysis:

3. I must disagree with part of the explanation given as to why group wasn't included as a random effect in one of the models. As I mention below, multilevel modeling (the use of grouping variables/random effects) isn't done in order to test predictions (though it can be used as such), it's about controlling for structure inherent in the data. Given that these animals exist in groups, then this fact needs to be accounted for in the statistical models. This seems especially important to me given that many of the models find variation with regard to group, which can be seen when investigating the summary output of these models using the code provided, and comparing the marginal vs. conditional r-squares. Furthermore, I don't understand this assertion of no expectation between groups, as there are at least two papers by the last author of this study arguing for the consideration of group-level variation in primate groups, especially when it comes to foraging:

Tournier, E., Tournier, V., van de Waal, E., Barrett, A. S., Brown, L., and Bshary, R. (2014). Differences in diet between six neighbouring groups of vervet monkeys. Ethology, 120(5), 471-482. https://doi.org/10.1111/eth.12218

van de Waal, E. (2018). On the neglected behavioural variation among neighbouring primate groups. Ethology, 340(10), 485-410. https://doi.org/10.1111/eth.12815

I think it is fine to say that the model wouldn't converge with group as a random effect and leave it there, as long as this is a limitation acknowledged in the results and their interpretation. Using a Bayesian approach would likely solve this issue because it performs better with smaller datasets and more complicated modeling than lmer, though I don't see it necessary that the authors change their statistical approach, though this is clearly the method preferred in the materials included supporting why group was dropped as a random effect in the R script.

4. When looking to the data and analysis code, there are 21 individuals that are measured twice in model 1 and 3 that are measured twice in model 2. It appears that some are males that moved groups, so I'm confused by the authors' reply that each dispersing male was only measured in the group that they first ate. Others appear to be animals that aged up during the study. I do see that if Individual is included as a random effect in models 1 and 2, the fit is singular. Perhaps a solution is to filter the observations to those in which the individual first ate, and mention this in the methods, as the authors replied they did (but my review of the analysis doesn't confirm). Another solution would be to run the models using a Bayesian approach, which would also fix the fitting problem with model 6 when including group as a random effect. Also, for model 2, there are 27 individuals marked as NAs for group AK19 for the measure of whether they ate in the first 4 exposures, when they have 0 for eating in the first exposure. Did I miss information as to why these animals were dropped from model 2 given they were measured in model 1? That is, while there are 191 observations (of 170 unique animals) in model 1, there are only 164 observations (of 161 unique animals) in model 2.

5. Model 3 doesn't seem to meet model assumptions for the residuals, and the R code state that this is fine for the sample size used. This needs to be reported in the main text-that model diagnostics reported an issue, and why the authors believe that this issue is not relevant.

6. The table for model 3 is not reported in the text.

Please see my detailed comments below:

20-21: Consider changing "according to the innovator" to "with respect to the initial innovator"

28-30: This sentence is a bit hard to follow. It might be better to talk about what was found, rather than what wasn't found, i.e.,

"Knowledgeable males and adults were more likely the targets of muzzle contacts compared to knowledgable females and juveniles, while also being less likely to be initiators."

48: What is meant by "potential" novel foods?

50-51: the line between "obtaining novel information" and "produces information or knowledge, potentially facilitating information" is murky. New information gained by an individual isn't always reduced by them…perhaps something like "and has the potential to produce information on which other group members can act." instead.

58: Behavioural patterns don't have be novel to be adaptive in new environments, as in the definition of innovation provides above-innovation could be a solution to a novel problem, including generalizing one behaviour, e.g., extractive foraging like getting acacia seeds from seed pods, to a novel situation, like extracting peanuts.

60: What's "this"? Behavioural plasticity? Behavioural patterns?

61: The risks association with novelty and innovation are unclear here. Up until this point, innovation and novelty have been largely been framed as positive, whereas only neophobia was mentioned with avoiding ris

66: Are juveniles required to innovate to learn about their environments? Learning about what is available as food in your group is not the same as finding a new food. Similarly, do dispersing animals need to innovate, or do they need to use previously-learned social skills to ingratiate themselves to a new group? Neither of these examples are a "a solution to a novel problem" or a "novel solution to an old one". Learning what to eat is not a novel problem…much like learning who your allies are isn't a novel one either. What really is the problem here is whether innovation happens at the level of the individual or group… certainly learning what to eat as a juvenile, or integrating into a new group, isn't a novel problem for this species, but one every animal must meet (save females re: dispersal). What I mean to say here, just because juveniles might be more prone to innovate, it's not necessarily from necessity, but could be a result of a developmental period that functions primarily to allow them to learn about their environments-while behavioural flexibility is necessary for innovation, it is not the same as innovation.

70: Why "nonetheless" here? The points following "nonetheless" seem to follow the points before it, and are not in contrast.

77: How is the risk diminished? Are captive animals are less neophobic because they are fed-is risk assessment, for which neophobia is a conserved trait across many species, is ontogenetically determined? Or is the point here that is more difficult to study such phenomenon in captivity, because there is no risk? I assume it's this latter point, but such risk assessment is never addressed in the current study, other than to mention that wild animals often experience changing environments, especially resulting from anthropogenic origins.

77-79: Individual differences with regard to innovation and behavioural plasticity has been shown to be true across many studies, including vervets, though there has been some conflicting evidence (cited below)…would it be better to say more work is needed, given the conflicting evidence?

Bono, A. E. J., Whiten, A., Schaik, C. V., Krützen, M., EichenBerger, F., Schnider, A., and van de Waal, E. (2018). Payoff-and sex-biased social learning interact in a wild primate population. Current Biology. Current Biology, 28(17), 2800-2805. https://doi.org/10.1016/j.cub.2018.06.015Bono, A. E. J., Whiten, A., Schaik, C. V., Krützen, M., EichenBerger, F., Schnider, A., and van de Waal, E. (2018). Payoff-and sex-biased social learning interact in a wild primate population. Current Biology. Current Biology, 28(17), 2800-2805. https://doi.org/10.1016/j.cub.2018.06.015

Renevey, N., Bshary, R., and van de Waal, E. (2013). Philopatric vervet monkey females are the focus of social attention rather independently of rank. Behaviour. Behaviour, 150(6), 599-615. https://doi.org/10.1163/1568539X-00003072

Canteloup, C., Hoppitt, W., and van de Waal, E. (2020). Wild primates copy higher-ranked individuals in a social transmission experiment. Nat Commun. Nat Commun, 11(1), 459-469. https://doi.org/10.1038/s41467-019-14209-8

79: "For example" might work better here rather than "moreover", since it's continuing the previous point. The reference to chimpanzees explicitly here is unnecessary, as the citations used include species beyond chimpanzees. "For example, across many species where males disperse, dispersing individuals…" would work better.

80: Citations [19] and [20] here use capuchins, not chimpanzees

85: Add "While these studies show that dispersing individuals…experimental (no "but")". However, at this point it is not clear to me why we need to compare multiple groups experimentally-this needs support.

70-100: This paragraph is very confusing to follow, as there are multiple independent points being made, including how social learning is beneficial, how the dispersing sex can import innovations or create them, a brief mention of the interface between social learning and innovation (i.e., it is implied that they are separate processes, but all of the benefits of this introduction point to species-level benefits of innovation, which require innovations to spread, so the brief mention of social learning here seems too minimal), and a discussion of social learning modalities.

97: Why the mention of social tolerance here? It is not clear how social tolerance speaks to the questions asked by this study.

103: What's "this"?

107: I'm still not sure why males need to innovate? Don't they, at most, need to generalize the behaviours of their previous groups to a new one? Why must they innovate?

123-124: Consider changing to "Our observations of innovation are limited in number, but further testing of the hypotheses we propose, as a result of our exploratory analysis that we present here, may aid our understanding of animal innovation.

125: Consider changing to "Given that animals learned to eat a novel food source, a behaviour that spread socially [24], (1a)…"

126-129: Why did you expect this?

130: "which" implies the results were known beforehand; "whether" might work better here.

131: Change "over all" to "overall"

131-133: Why differentiate across exposures here, especially since the predictions are the same? I know there are 2 different models, but I need to know why here so that I can understand the differing predictions. Moving the explanation as to why the first 4 exposures were considered to here would be helpful.

132: What's "this"?

134-136: There seems to be a bit of double-dipping here, as the findings from one dataset (at least for 2 groups) are used as evidence for a prediction for data in the same dataset (the current study). Additionally [24] and [39] found that higher-rankers are more likely to be observed, not that they were more likely to uptake a novel food. In fact, the common prediction here is that higher-ranking animals should be more neophobic, because they have better access to food and thus eating unknown food is riskier for them given their prime access to food overall:

Wolf, M., van Doorn, G. S., Leimar, O., and Weissing, F. J. (2007). Life-history trade-offs favour the evolution of animal personalities. Nature. Nature, 447(7144), 581-584. https://doi.org/10.1038/nature05835

Greenberg, R. (2003). The Role of Neophobia and Neophilia in the Development of Innovative Behaviour of Birds Animal Innovation. In S. M. Reader and K. N. Laland (Eds.), Animal Innovation (pp. 175-196). Oxford University Press. https://doi.org/10.1093/acprof:oso/9780198526223.003.0008

Laland, K. N., and Reader, S. M. (1999). Foraging innovation in the guppy. Animal Behaviour. Animal Behaviour, 57(2), 331-340. https://doi.org/10.1006/anbe.1998.0967

But see:

Amici, F., Widdig, A., MacIntosh, A. J. J., Francés, V. B., Castellano-Navarro, A., Caicoya, A. L., Karimullah, K., Maulany, R. I., Ngakan, P. O., and Hamzah, A. S. (2020). Dominance style only partially predicts differences in neophobia and social tolerance over food in four macaque species. Scientific reports. Scientific reports, 10(1), 1-10. https://doi.org/10.1038/s41598-020-79246-6

Drea, C. M. (1998). Social context affects how rhesus monkeys explore their environment. American journal of primatology. American journal of primatology, 44(3), 205-214. https://doi.org/10.1002/(SICI)1098-2345(1998)44:3%3C205::AID-AJP3%3E3.0.CO;2-%23

140-142: This makes sense to me, but I have no idea why this prediction is made. Perhaps moving the explanation given to the me

146-147: "the media" implies that this is how animals are learning to eat the peanuts…but the author replies to reviewers mention multiple times that this is not what is meant.

147-150: This has been previously found in [31]; thus there is both theoretical and empirical support for this prediction.

152: [31] found this, and hypothesized that social tolerance is necessary for muzzle contact to afford foraging information, but perhaps and additional citation here about how lower ranking animals are tolerated by fewer group members would help make the point.

153-154: Initiated vs targeted…via muzzle contact?

158-161: I don't follow…this seems to assume that initiators are seeking information and this seeking will outweigh any social tolerance constraint, but only previous study of muzzle in vervets found that tolerance was the best predictor of the behaviour. Thus, this prediction needs more support as to why it is in the opposite direction of what the literature shows, i.e., that social tolerance constrains information-seeking and information spread, akin to Carter's (2016) sequential social learning hypothesis.

161-173: What kind of different experiences of novelty arise from the life history trajectories of the philopatric vs. dispersing sex? Again, why do dispersing animals experience more novelty? Why should we expect the groups to which they are dispersing to have significantly different diets that we can call "novel"? Do dispersing animals need to gain totally new information? Surely not, as the kinds of foods available are likely very similar and behavioural generalization can do a lot of work. When it comes to conspecifics, isn't a plausible alternative hypothesis that dispersing animals need to enter groups using the same skills needed to integrate in to the adult social networks as they age in their natal groups before dispersal…so what counts as novel here? Again, I see these problems as being neither novel or the success of dispersing animals after integration into new groups as being dependent on a novel solution. It seems to be the novelty of interest here is much larger, as mentioned at the beginning with reference to anthropogenic-induced changing environments, rather than the kinds of problems these animals have encountered throughout their evolution.

164-165: Why this prediction? I can think of some reasons why this is, but there is no support for this prediction that naive adults should initiate and receive at the same rates…prior evidence [31] suggests that adults should more often be targeted than initiate, so it seems here that this prediction relies on explicit knowledge seeking, which requires the prediction the muzzle contact is primarily used to gain novel information.

166-167: Why should females initiate if they don't "need" info, which is what this prediction is implying…presented like this, this prediction reads as if the results were already known when it was made.

169-171: Why would malesy stop initiating? Why does initiating influence receiving? One does not preclude the other…

170-173: I don't follow this prediction at all… that knowledgeable males are somehow more tolerated…doesn't the work reviewed in the introduction at least imply that new immigrants have novel information, and by definition, new immigrants are less known to the group, so should be interacted with less. How does a muzzle contact initiator know that a new male is knowledgeable? And why wouldn't a new male be less tolerated by others compared to an established male, who has relationships with the group? Again, I'm not sure of the dismissal of tolerance here, when the only previous work on muzzle contact in vervets found that social tolerance, above all else, influences muzzle contact behaviour? Especially given that prediction 2b makes a social tolerance prediction, that lower ranking animals will initiate less than higher ranking animals.

179-182: Why mention this here? Perhaps this would be better at the very beginning of the results, or near where the differentiation is first referred to in the results.

209: Should this be "his" instead of "their"?

213-221: This use of uptake is confusing… in the reply, the authors state " the reason we talk about 'uptake' rather than social learning is that we really see this as a case of social disinhibition of neophobia, rather than more detailed social learning such as copying or imitation" but this disinhibition hypothesis is never mentioned in the introduction. The introduction needs to make clear this distinction, and why, despite that this behaviour was previously shown to be socially transmitted, social learning language here. I see no reason not to report that this behaviour is socially transmitted and that this study takes the opportunity to explore who innovated and whether socio-demographic variation corresponded with innovation, as well as the opportunity to further explore muzzle contact as a means of learning about novel foods given previous evidence showing that muzzle has the potential for being a learning modality, rather than proposing an entirely different mechanism.

Also, how does one prove the difference between the uptake of the innovation being the result of social disinhibition and the topography of opening the peanut being socially transmitted? I understand the use of EWA to show the latter, but am not sure how that is separate in fact from the former…how does one show the approach and willingness to interact is only socially facilitated, but the opening itself is socially learned? Especially given that all of the results in this study are presented in regard to who extracted and ate the peanuts, and not some other measure of neophobia.

218-220: Wasn't rank standardized, with 0 being the highest ranking? Because this model found a positive "non-significant trend" of rank, doesn't this mean that lower ranking animals (e.g., those with higher values in the model) were more likely to eat at first exposure? And this is the same for the findings over 4 exposures (225-228), as well as frequency of initiating muzzle contacts (lines 253-260), and of being targets (lines 267-269).

219-221, 267, 268: the use of "trend" when results are not significant has been recently been convincingly objected to by Wood, Freemantle, and Nazareth:

https://www.bmj.com/content/348/bmj.g2215

I'm not sure interpreting the direction of rank effects is useful here given that the rank variable did not meet the significance threshold used. Here is the place where the use of a Bayesian approach would allow such interpretations, as Bayesian credible intervals can be interpreted in this way, whereas p-values cannot (more on a Bayesian approach below).

219-onward: Was R-squared calculated for any of the models? This would help in understanding how much variance in the data each model explained. Additionally, the n of each model should be reported.

295-296: Shouldn't this be low ranking ate the novel food more readily?

297: This seems to me to be a question of what counts as exposure…if, as olfactory contact with novel food increased is considered the actual measure of exposure relevant to muzzle contact, than the number of animals eating the food is just a proxy for this, i.e., it isn't about the number of animals eating the food, but the proportion who have had olfactory experience. Thus, as this proportion of olfactory exposure increase, muzzle contact decreased.

316-317: Why would you expect this?

329-332: What about Boc doesn't meet these criteria?

307-340: This is very good, and would help them in the introduction!

364: Does this need to be re-interpreted given that the rank effects in the results are presented in the opposite direction (i.e., a positive effect in the model represents lower ranking animals engaging more in the target response than higher ranking animals)

370: These results are impossible to interpret given that the random effects are not reported.

404-407: This needs to be reevaluated given that it was actually lower ranking that ate more; however, if this was the finding, a discussion as the fact that this is in contrast to the literature that predicts that higher ranking animals should be more neophobic is warranted.

431: Same point of the rank interpretation

444: This is also in line with [31], which is on vervet monkeys

446: as in 29, 30, and 31

448: Could it be that adults were acquiring information from juveniles, but not applying it, for some reason, akin to Carter et al. (2016)?

456: Again, this needs to be reevaluated-lower ranking animals were more knowledgeable, and were more often the targets of muzzle contact; [31] found that lower ranking animals were more likely the targets of muzzle contact, and used social tolerance to understand this; it's not that low ranking receivers are less tolerated by higher ranking initiators, it is that lower ranking animals cannot refuse initiators as much as higher ranking animals might. (See figure 2b and table 4 of 31; in this paper, higher values indicate higher rank, so the negative effect of rank in table 4 indicates that lower ranking animals were more often the targets of muzzle contact).

466-487: This is great, and an important contribution. Context matters for behaviour, but it is rarely explored-this neglect may be an important reason why social learning isn't as widespread as we'd expect (in my opinion).

510-516: The interpretation of males needs more work…why would males remain bold, outside of dispersal, when it is so risky, and arguably when they have established relationships with group members? And this interpretation is in contrast to the discussion in 307-340 that recent immigrant males are in a specific state that makes them more likely to innovate.

615-618: Why wasn't exposure time included as a control in the model, given that it varies?

618-622: This is in contrast to 595-598: was it always after sunrise in the sleep site, or opportunistically?

660 onward: It would be helpful to mention model names with each description, i.e., that 1b first exposure was model 1, 1b 4 exposures was model 2, etc.

Table 3: Where are the group effects reported, i.e., the random effects?

673-675: This needs to be mentioned much earlier (see my comments questioning why 4 exposures above)

677-679: Why is this prediction here, and not in the introduction? And what is the support for this prediction?

691-696: Again, why are the predictions being restated and/or elaborated in the methods? Perhaps it would be easier to number the predictions in the introduction and refer to them here.

699-701: What were the offsets for these models, given they were poissons?

727: But this manuscript presents many predictions for rank, finding a rank effect in many of the models. I don't see a reason for dropping rank here.

733-736: Please see my main comment above regarding the use of group as a random effect.

"Final model" is used throughout the manuscript (e.g., lines 273-274, 738), implying the authors used a model comparison approach, but any information about how models were selected is not provided.

*Reviewer #2 (Recommendations for the authors):*

I commend the authors for their hard work in improving their manuscript to accommodate the comments raised by myself and the other reviewers. However, I still feel there is considerable conceptual fuzziness that constrains a clear interpretation of the data presented here, as well as some remaining issues with the analysis. Much of this is made apparent in the authors' Reply to Review, so I will primarily address this. Below that, I have some more minor comments on the revised manuscript.

1) Conceptual and inferential ambiguity

"My comment: Line 281: More detail needed. Did these knowledgeable individuals typically have their mouths full of the target food during these events? If so then it seems parsimonious to assume the muzzlers were simply following this rather than tracking knowledge-states.

Authors reply: We do not claim that they track knowledge states – we are claiming that they can tell who is currently eating or has eaten a food that they do not know about, and try to obtain information about that food. We use the word "knowledgeable" for our human readers to easily identify and refer to "individuals that have already learned to extract and eat peanuts". We never report in the manuscript that we are inferring that the monkeys track the knowledge state. We do assume that if they are close enough to muzzle contact, they are close enough to have probably seen them eat the food."

"…we never report in the manuscript that we are inferring that the monkeys track the knowledge state." Throughout the manuscript the authors make statements to this effect…"

I'm particularly surprised by this final comment since one need not even read past the abstract to see that it is clearly untrue: "Finally, knowledge influenced females and juveniles less than males and adults in becoming more likely targets than initiators.". The manuscript is riddled throughout with examples of such causal language that heavily implies a direct effect of knowledge on the outcome measures. This is extremely misleading and serves no purpose. The word 'knowledge' should be removed from the manuscript entirely and the authors find another way to describe their variable. For example, why not just call the 'knowledgeable' individuals "demonstrators"?

Below I answer several comments at once:

"We did not intend to claim that muzzle contact was the specific mechanism by which individuals learned to extract and eat peanuts – we rather use this experiment to evaluate the function of muzzle contact in the presence of a novel food."

"For this, and the above points: We did not record an observation network for the groups added in this study and are not able to answer this – it is not the focus of this study. For this reason, we do not make claims in this line in the present study, and are cautious with our social learning related language. Whilst we examine the role of muzzle contact in acquiring information about a novel food, we do not expect this behaviour to be a necessary prerequisite in being able to extract and eat this food – indeed many individuals who learned to eat did not perform muzzle contacts. This aspect of the study is about using this novel food situation to explore whether muzzle contact serves information acquisition – which our evidence suggests it does. Moreover, the processing of this food is not complex and is similar to natural foods in their environment, and we do expect individuals to be capable of reinventing it easily (and this point with Tennie's hypothesis is actually discussed in Canteloup et al. 2021 paper) – but the point here is that their natural tendency is to be neophobic to unknown food, and therefore they do not readily eat it until they see a conspecific doing so, after which they do. And we also used this opportunity, though in a very small sample size, to investigate which individuals would overcome that neophobia and be the first to eat successfully."

"See above – the reason we talk about 'uptake' rather than social learning is that we really see this as a case of social disinhibition of neophobia, rather than more detailed social learning such as copying or imitation, as it would be in a tool-use setting, for example (though in Canteloup et al. 2021 paper, evidence is found that the specific methods to open peanuts are socially transmitted)."

"…there is a distinction between information acquisition and information use – obtaining olfactory information about a novel resource that conspecifics are eating is not the same as learning a complex tool use behaviour for which detailed observation of a model is required. We are not claiming that muzzle contact is THE mechanism by which the monkeys learn how to eat the food"

To summarise: When I suggested the authors have implied a role in social learning, they deny this (okay! But I'm unsure about the need for evasiveness on this one – there are more kinds of social learning than just action-copying). Nevertheless, they argue that the monkey are 'gaining information' about the food and that the decline in MC as they become more knowledgeable implies a role in learning (social or asocial) or 'overcoming neophobia'. This seems plausible and a worthy hypothesis to test!

However, when I asked for evidence that individuals who MC more often are more likely to learn how to eat the food, the authors refused to examine this on the basis that "MC is not THE mechanism by which learning occurs". Regardless of whether it is THE mechanism, or simply a means of overcoming neophobia, if MC serves the function the authors have argued then it should lead to an increase in the likelihood or rate of uptake – otherwise what is the point? The authors refusal to support their argument with easily accessible data (they have apparently already recorded the identity of all individuals and their feeding/Mc behaviour) that would robustly confirm the behavioural function one way or the other is quite frustrating.

In fact, the authors do present some data that contradicts their hypothesis:

Line 681: "Inspection of Figures 4A and 4D suggests that juveniles, relative to adults, still initiate more than they are targeted even when knowledgeable."

Why should knowledgeable individuals muzzle-contact at all? These individuals already have the information they need. This is a major hole in the authors' argument.

"We recorded muzzle contacts visible within 2m of the box, so individuals were not necessarily eating at the box at the time of engaging in muzzle contacts. However, the majority of muzzle contacts that we could record took place directly at the edge of the box – at the location where the food is accessed – so an individual would not likely be if they were not able to have access to the food. It is possible they could be there and not eating, but they would not have been chased off, otherwise they would not be able to engage in muzzle contacts there. But it is not entirely clear what the reviewer's point is here."

If muzzle contact was only recorded within 2m of the food source, is it any wonder that knowledgeable individuals were chosen more often? Surely the majority of individuals at the food are those who have figured out how to eat it. See the comment below this one.

"My comment: What proportion of PRESENT (not total) individuals were naïve and knowledgeable in each group for each trial (if 90% present were knowledgeable, then it is not surprising that they would be targeted more often)?

Authors reply: We agree somewhat with this statement, but given the multiple ways we show the effect of knowledge – both at the individual level and the group level (effect of exposure number i.e. overall group familiarity) – we feel we present enough evidence to establish the link between knowledge of the food and muzzle contacts. We find that the model showing the interaction between exposure number and number of monkeys eating on the overall rate of muzzle contacts actually addresses this issue, because we see that when many monkeys are eating during later exposures when many were indeed knowledgeable, the rate of muzzle contacts is massively decreased. Moreover, if 90% of the individuals present are knowledgeable, then only 10% of the individuals present are naïve, and we show both that knowledgeable individuals are targeted, but also that naïve individuals are initiators."

The authors have not really addressed my original point here, so I apologise if it was unclear. First, I accept the authors' conclusion that knowledgeable individuals are less likely to carry out a MC (but see below for problems regarding their interpretation of this). Instead, I was raising a point of basic sampling bias and statistical inference: If the majority of individuals at a feeding site are knowledgeable, then even a blindfolded individual who is choosing recipients are absolute random will select knowledgeable individuals more frequently. If all of the knowledgeable individuals are male, a blindfolded individual will similarly demonstrate a "bias" towards male, knowledgeable individuals. If this is not factored into the analysis then it is not inferentially sound.

"…but we do believe that the clear separation between naïve individuals initiating and knowledgeable individuals being target, and the decrease of the rate of this behaviour as groups' familiarity with the food increases – is good evidence that this behaviour functions to acquire information about a novel food."

That is one interpretation (but see comment above re: sampling bias for initiators) – Another explanation is that these behaviours are simply mutually exclusive at a given moment in time: once they know how to eat the food, they prefer to spend their time doing this than engaging in MC behaviour. Rates of resting, grooming, etc within 2m of the food presumably also decrease once the monkeys have figured out how to eat it, not because there is any causal relationship between these behaviours but because they can only do one thing at a time and feeding is a priority.

2) Analysis

The authors have heavily revised their original analysis and it is largely improved. I have a few remaining issues which I describe below.

"My comment: The text for this muzzle-contact analysis would indicate that this model was not fit with any random effects, which would be extremely concerning. However, having checked the R code which the authors provided, I see that Individual has been fit as a random effect. This should be mentioned in the manuscript. I would also strongly recommend fitting Group (it was an RE in the previous models, oddly) and potentially exposure number as well.

Author reply: The model about muzzle contact rate never contained individual as a random effect because individuals are not relevant in this model – it is the number of muzzle contacts occurring during each exposure. However, the reviewer might refer here to the model that we forgot to provide the script for. Nonetheless, we have substantially revised this model, it now (Model 3) includes all groups, and has group as a random effect."

I do not accept that individual is not a relevant random effect. I understand that the model is intended to examine group-level rates of M-C, but groups are made of individuals. Let us imagine a scenario where a single individual is a highly prolific muzzle-contacter in group BD, accounting for 95% of M-C events, and NH contains no such individuals. An analysis that takes a straightforward group rate without accounting for individual contributions will likely find a significant difference between the two, driven by a single individual. If the authors have structured their data and analysis in such a way that they cannot control for this factor then that is an issue. One "quick and dirty" solution, that would require a minimal amount of restructuring of the data, would be to take an individual rate for each monkey in a group, or at the feeding site, or whatever, and then derive the group average from this. Otherwise, it is not clear what we can infer from this analysis.

"Authors: We have now checked for overfitting in our models."

Where is the evidence of this, please? There are metrics and methods that can be used to achieve this (such as AIC/LOO-based model comparison approaches I suggested in my last review) but the authors do not report them.

"We included individual as a random effect, but we did not include group as a random effect here for two reasons. First, we did not have any theoretical basis to expect residing in different groups to have an effect here, since we were concerned with the effects of life history strategies of individuals on their information acquisition behaviour, which should not differ for individuals from different groups."

This is not theoretically sound. Individuals from groups are more likely to be similar than individuals from different groups – this is the purpose of grouping variables. They live in similar ecologies, share life history events, and are more closely related.

[Editors' note: further revisions were suggested prior to acceptance, as described below.]

Thank you for resubmitting your work entitled "Role of immigrant males and muzzle contacts in the uptake of a novel food by wild vervet monkeys" for further consideration by *eLife*. Your revised article has been evaluated by George Perry (Senior Editor) and a Reviewing Editor.

The edits to the manuscript were much appreciated but unfortunately have also brought to our attention some additional issues with your statistical analysis that must be addressed, as outlined below.

1. The issue is that once you reported your dispersion parameter results, it is now clear that Models 4 and 5 are highly underdispersed, and model 3 moderately so. Underdispersion can be considered as much an issue as overdispersion for poisson models so we urge you to rethink the error structure used for these models so that you do not violate the assumptions of a poisson distribution.

---

## [Author Response]

Essential revisions:All three reviewers found this study to hold great potential in providing significant new insights into the field of social learning and transmission in wild animals. However, due to a number of important concerns with the analysis it is difficult to ascertain whether the authors' claims are valid given the evidence. We invite the authors to address these major concerns in a substantially revised version of the manuscript.1) Statistical analysis needs revising:A number of concerns regarding multiple testing and the structure of your mixed models require attention. In particular, please consider using a multimodel approach due to the exploratory nature of your analyses (see suggestions from Reviewer 1 and 2) and revise the structure of your mixed models to include essential random effects where necessary, and address potential confounding variables such as group size, combining age and sex into one variable, and the directionality of muzzle-muzzle contact initiations (see all 3 Reviewer comments for details). Please also ensure the code for all your models/analyses have been provided.

We have updated our models according to the reviewers’ comments. Importantly, we added two new models in the muzzle contact section, due to reviewer #3’s correction about the model we originally used to answer question 3b (Individual variation in muzzle contact behaviour). They correctly noticed that the model we used was not suitable for some of the comparisons we were drawing from it. We retained this model (now Model 6) as the things it does show are very interesting regarding the life history strategies of the two sexes in this species, and still strengthen the overall message of the paper. The two new models that we have added are suitable to examine what we originally intended, which was the likelihood of (a) initiating muzzle contacts (Model 4), and (b) being targets of muzzle contacts (Model 5), between different ages, sexes, ranks and knowledge-bases. We are very grateful for that correction.

Second, following reviewer #1’s advice, we checked all models for whether individual needed to be a random effect, but our original models for which they made this comment only include each dispersing male once – in the group where they ate first – so there was no repetition of individuals and we did not need to add this there. Regarding group as a random effect, we now include this everywhere that we believe it is necessary – which is all but one model (Model 6) where we did not, because we do not theoretically expect residing in different groups to have an effect there, and the model was too complex to include it – we mention this again below, and we also show our process in the R code.

Lastly, whilst model selection approaches were recommended, we chose not to implement this, because wherever we used models, we did have specific predictions based on previous work and theory. The only time we actually referred to exploratory work is for what were originally questions 1 and 2a, and now we have now merged into the present question 1. We do not formally analyse these data using models. We hope this is clearer now from the way we have now written the hypotheses and predictions (also following advice from reviewer #2).

2) Methods need to be more transparent:All reviewers found that parts of your study lacked sufficient detail to be repeated by others. Could you please provide clear criteria for the various decisions made throughout the study as well as justifications for cut offs used (e.g., why 3 months for immigrant males?). The rank calculation also needs more clarity and various decisions made by the authors are not justified/clear. Please also provide interobserver reliability tests regarding coding (see Reviewer 1 and 2 for details).

We have now addressed these issues: more detailed information regarding our observations of the tenure of the innovators and other relatively short-tenured males (lines 312 – 317); reported interobserver reliability for the video coding (lines 647 – 651); explained the criteria for ending an exposure (lines 614 – 615); and other methodological clarifications are indicated below next to the relevant comments from the reviewers.

3) Study needs reframing:All three reviewers found the introduction lacked direction and conceptual clarity. Please provide a more thorough rationale for your study and integrate this into a list of explicit research questions and predictions. The discussion would also benefit from consideration of alternative explanations (see details in comments from all 3 Reviewers).

We have added more background supporting our interest in dispersers in the introduction, and completely re-wrote the research aims with much more detail and supporting evidence to back up our hypotheses (lines 125-173). We also add more discussion of alternative hypotheses to the discussion (lines 370 – 380 regarding uptake of novel food; and lines 412 to 417, regarding muzzle contact results). We hope this is clearer now.

Reviewer #1 (Recommendations for the authors):I have some suggestions regarding the methods:– It needs to be reported what decided when a trial was begun, and ended, as this would help explain the differences in the trial lengths.

This is now specified in the method (lines 614-625).

– It would be very helpful to report how STRANGE these animals are, especially given this manuscript is about innovation; see:Webster, M. M., and Rutz, C. (2020). How strange are your study animals. Nature. Nature, 582, 337-340. https://www.nature.com/articles/d41586-020-01751-5?sf235295265=1Farrar, B. G., and Ostojić, L. (2020). It's not just the animals that are strange. Learn Behav. Learn Behav. https://doi.org/10.3758/s13420-020-00442-5

We now refer to the STRANGE framework when discussing about potential group size and experimental history effects (lines 371-380).

I have some suggestions regarding the analyses:– The R package DHARMa is a great resource for model residual diagnostics.

We thank the reviewer and we used this.

– There are no effect sizes reported for any of the models.

We now report effect sizes of our models.

– The code for the first muzzle contact model, looking at rate, wasn't included, and so I was unable to review it. Further, it was unclear as to if only a subset of the available data was used for this model (groups with lots of eaters), and if so, why. Including group as a random effect could help account for any group differences that the authors may have felt relevant to subsetting the data, thus allowing all of the data to be examined.

Apologies for that omission – it is added now, and all groups are in.

– Collecting data in the field is quite different from coding from videos, and reporting a reliability measure on the video data would help readers to assess the manuscript's findings.

We have added this.

– It seems to me that all models need both ID and group as grouping variables (random effects). This is because all of the models, as far as I can tell, include some of the same adult males (though on different troops), and all of the analyses are all conducted across all of the groups, requiring group to be a random effect.

ID was already a random effect in all the models with repeated rows of data per individuals, but we actually only included the dispersing males in groups where they ate first, so this was not an issue. Group is now a random effect everywhere where we can see a theoretical reason to include it – we did not include it in the final muzzle contact model because the model was too complex to converge with it in, and we also see in this case, no reason for the groups to have different outcomes. This question is more related to individuals’ life history differences, specifically between age and sex classes, and we do not expect that be variable among groups.

– Exposure to the experiments varied widely across groups, and it is unclear if all animals on the same groups were around for the trials (I'm assuming they weren't as some trials were dropped due to low numbers of participants). These discrepancies need to be accounted for in the manuscript and analyses.

We no longer drop these exposures. We are unable to obtain accurate records of how much of the group was present from our videos, but in all except the originally dropped exposures, the whole groups were in the area and within visual access of the experiment at least at some point, and therefore had to opportunity to approach the box if they liked (besides intragroup dynamics e.g. monopoly by high rankers)

– There is no explanation as to why there are three different versions for the provided models looking at muzzle contact (1 model with (non-normalized) rank, 1 with two interactions but without rank, and 1 with three interactions, but also without rank). Why was rank dropped? Why were interactions included? If multiple models are going to be considered, a model comparison value, like AIC, should be included to compare the models (after first explaining the theoretical reasons as to why different versions were considered).

We have updated this model and how we set it up, also in accordance with the comments from Reviewer #3 who correctly found discrepancy between what we intended this model to do, and what it actually did. Nonetheless we found what it did measure to be highly interesting and have reframed it, whilst adding appropriate models for our original intention. This issue with rank is no longer relevant as we did not expect rank to have an effect in what was actually being measured here.

– How was rank normalized, and why was normalized rank considered in some models but not others?

This was a mistake and we apologize for that. Has been checked and all models now presented which test rank use the normalised rank. It was normalised (or maybe “standardised” is better terminology) so that despite group size, all ranks fall between 0 (highest rank) and 1 (lowest rank). [(rank -1) / group size] – this shows us the proportion of the group that each individual outranks.

– Figure 2A y-axis is unclear- shouldn't it be closer to 50% for BD given 35 innovated? Or is this just after the first exposure? Or first eating event?

The y-axis label and the figure caption do both specify the first eating event.

Other questions/ suggestions:– Line101: why the possessive on "groups"?– Line 286: this is in line with the hypothesis of Nord et al. 2020, which concluded that, "both kin and low-rank- ing animals serve as discriminative stimuli for social tolerance and that foraging animals serve as discriminative stimuli for food availability" Though, this manuscripts first muzzle contact model (for which the code was not provided) found no evidence of a rank effect (though rank wasn't included in the interaction models), which is in contrast to Nord et al., thus providing an interesting extension to findings about muzzle contacts, in that social tolerance may play less of a role in cases of novel foods.

Our results support Nord et al. results now, and we refer to this study (ref N°31) in the discussion. The previous difference was due to our mistake of using non-standardised rank in that model (line 459).

– Line 289: This also agrees with Nord et al.– Line 300-302: Do the juveniles referred to here out rank adults? Are ranks calculated across all age types in these data?

Ranks are calculated across all adults and juveniles, and infants are given the rank just below their mother (due to lack of data on infants) – this is described in the methods now.

– Line 303-305: The manuscript reports that juveniles have information, but aren't passing it…how is this an adaptation to risky behavior, if juveniles are more likely to eat something novel anyway? Isn't this doubly bad for juveniles, in that they are more likely to eat something they don't know, which can be dangerous, and also aren't functioning as a source of information when they have it?

We think it could be adaptive that individuals do not learn from juveniles, because if they are more prone to risky behaviours, it is better that these risky behaviours do not spread in the group (lines 398-401) – the adaptive aspect is that by preferring to learn from adults, hopefully the information that is learned is more reliable as they are more risk-averse. However, with our new modelling approach, we did not find a difference in how adults and juveniles were targeted with muzzle contacts, though it is what we expected, and we discuss this in lines 448-452.

– 309-311: This could possible be tested putting age and sex, as a combined variable, in the model, creating no need to have an interaction between age and sex and knowledge, and instead would only need one between Age-Sex and knowledge

We tried this but it did not change the results, and made the results more complicated to discuss. We maintain the separate interactions.

[Editors' note: further revisions were suggested prior to acceptance, as described below.]

The manuscript has been greatly improved but there are some remaining issues that need to be addressed. In particular, the responses to reviewer concerns regarding the conceptual framing of the study and the analysis are not sufficiently addressed and we find the justification for not making the suggested changes unsatisfactory.In your revised submission please ensure you adequately address the major comments provided by Reviewers 1 (points 1-6) and 2 (points 1 and 2).To summarize, the introduction still requires substantial revisions to provide a more clear conceptual framework and predictions, along with how this aligns with your analyses. The analysis also requires attention as key components of models are still missing, (i.e., random effects) and the transparency of your results will be much improved if sample sizes and R-squared values are reported for each model. See Reviewer comments below for details.Reviewer #1 (Recommendations for the authors):The authors have done a great job making the introduction more applicable to the research conducted. It is still a bit disorganized, which I comment on explicitly below. Furthermore, the exploration of role of knowledge in information-seeking is an important contribution, as this kind of social/contextual account is not often a topic of study in the social learning and innovation literature.1. The introduction seems to present the idea that learning what to eat is innovation, and that dispersing is reflective of innovative abilities, though I'm not certain that this is what the authors intend. In the discussion, the explanation as to why dispersing animals may be more likely to innovate when they are dispersing (a "transitory exploratory behavioural syndrome") is much more clear. Indeed, it might work better to introduce this idea in the introduction, and continue along the same lines with regard to juveniles-that given that they need to learn about their environments, they might be better primed to innovate akin to the transitory behavioural syndrome of dispersing animals, rather than viewing ontogeny as a constant state of innovation. However, it is still unclear as to when males are considered immigrants vs. residents. Surely after group integration, they are no longer experiencing a "transitory exploratory behavioural syndrome.". Explicitly outlining this distinction (dispersing vs. resident) would greatly help the manuscript.

This is what we intended. We added explanation of our ‘transitory exploratory behavioural syndrome’ hypothesis to the introduction (L 71-72).

2. Additionally, it is very odd that all of the predictions presented in the introduction are exactly what is found, even when they are in contrast to the literature. This is especially difficult given the predictions about rank, and that, as long as I'm reading the methods correctly, the rank results are actually presented backwards, meaning that the authors found the opposite of what they report (i.e., because increasing rank values equals decreasing rank--higher ranking animals are represented by lower values--and the models found a positive effect of rank, then it is that lower ranking animals were more likely to eat the novel foods, not higher ranking animals; this is in line with the neophobia literature that predicts that higher ranking animals should be more neophobic, in contrast with the prediction provided in the introduction).

Whilst the reviewer has previously spotted other errors in our scripts and data files, for which we are very grateful, in this case they are mistaken. We have checked and we report the rank results correctly, and describe them in the text accurately. The effects of rank are not positive – they are negative – odd ratios <1 are negative. To make this clearer, we have added a column to the results table (L850) that also shows the model coefficient, as we believe the reviewer has mistakenly read our odds ratios as coefficients, thus believing them to portray positive effects, when they are in fact odd ratios portraying negative effects.

I have a number of questions with regard to the analysis:3. I must disagree with part of the explanation given as to why group wasn't included as a random effect in one of the models. As I mention below, multilevel modeling (the use of grouping variables/random effects) isn't done in order to test predictions (though it can be used as such), it's about controlling for structure inherent in the data. Given that these animals exist in groups, then this fact needs to be accounted for in the statistical models. This seems especially important to me given that many of the models find variation with regard to group, which can be seen when investigating the summary output of these models using the code provided, and comparing the marginal vs. conditional r-squares. Furthermore, I don't understand this assertion of no expectation between groups, as there are at least two papers by the last author of this study arguing for the consideration of group-level variation in primate groups, especially when it comes to foraging:

Group is now included as random effect in all the models. We provide the variance and standard deviation of the random effects and the marginal and conditional r-squares in table 4.

Tournier, E., Tournier, V., van de Waal, E., Barrett, A. S., Brown, L., and Bshary, R. (2014). Differences in diet between six neighbouring groups of vervet monkeys. Ethology, 120(5), 471-482. https://doi.org/10.1111/eth.12218van de Waal, E. (2018). On the neglected behavioural variation among neighbouring primate groups. Ethology, 340(10), 485-410. https://doi.org/10.1111/eth.12815I think it is fine to say that the model wouldn't converge with group as a random effect and leave it there, as long as this is a limitation acknowledged in the results and their interpretation. Using a Bayesian approach would likely solve this issue because it performs better with smaller datasets and more complicated modeling than lmer, though I don't see it necessary that the authors change their statistical approach, though this is clearly the method preferred in the materials included supporting why group was dropped as a random effect in the R script.

We have decided to remove model 6 from our analyses. Given that it was originally intended to investigate what models 4 and 5 now correctly investigate, it is not worth the small amount of additional information that it might glean, especially given that there are too many problems with the model diagnostics and model convergence.

4. When looking to the data and analysis code, there are 21 individuals that are measured twice in model 1 and 3 that are measured twice in model 2. It appears that some are males that moved groups, so I'm confused by the authors' reply that each dispersing male was only measured in the group that they first ate. Others appear to be animals that aged up during the study. I do see that if Individual is included as a random effect in models 1 and 2, the fit is singular. Perhaps a solution is to filter the observations to those in which the individual first ate, and mention this in the methods, as the authors replied they did (but my review of the analysis doesn't confirm). Another solution would be to run the models using a Bayesian approach, which would also fix the fitting problem with model 6 when including group as a random effect. Also, for model 2, there are 27 individuals marked as NAs for group AK19 for the measure of whether they ate in the first 4 exposures, when they have 0 for eating in the first exposure. Did I miss information as to why these animals were dropped from model 2 given they were measured in model 1? That is, while there are 191 observations (of 170 unique animals) in model 1, there are only 164 observations (of 161 unique animals) in model 2.

It seems that the reviewer may not have overlooked our explanation that AK19 only had one exposure in 2019 (in the methods: line 523), which is why they were not given scores for eating across all four exposures (and thus absent from model 2). This is why they have NAs for eating across all four exposures. As we described in the text (L174; and now also in the methods in L 523), we considered AK in 2019 and 2020 separately because 40% of the group changed and there was a year worth of gap between the single exposure in 2019 and the four subsequent ones in 2020. As no individuals had eaten in 2019, the peanuts were still novel to them in 2020 as a food source. Nonetheless, this does not actually affect any of the analyses (see next paragraph) and is just a descriptive difference. The issues you noted with the numbers of individuals should now be rectified, with dispersing males only considered in the group they were first present in, and no repeated individuals in the actual analyses.

Part of this issue arose as we intended to look at the exposure with the first eating event but had accidentally used the first exposure. We have now corrected this data file in accordance with looking at the exposure with the *first eating event* (see L215 in the results and L 555 in the methods). Therefore, the AK19 individuals are not in either analysis, only AK20 are because they had both a first eating event and four exposures. We could remove AK19 all together from the data file but have not done so for completeness of the data set, and to allow analysis of the very first exposure to be carried out if wanted.

5. Model 3 doesn't seem to meet model assumptions for the residuals, and the R code state that this is fine for the sample size used. This needs to be reported in the main text-that model diagnostics reported an issue, and why the authors believe that this issue is not relevant.

We have completely re-done this model in accordance with reviewers 1 and 2 comments about it – please see L233-244 in the results and L 608-612 in the methods as well as Table 3. Model diagnostics are good as one can see in the model script.

6. The table for model 3 is not reported in the text.

We had previously reported this model in the text, and thus not repeated its results in the table because it was a linear model not a glmm like the others presented in the table. Since we changed this analysis to a glmm we now report model 3 in the table.

Please see my detailed comments below:20-21: Consider changing "according to the innovator" to "with respect to the initial innovator"

We have changed this.

28-30: This sentence is a bit hard to follow. It might be better to talk about what was found, rather than what wasn't found, i.e.,"Knowledgeable males and adults were more likely the targets of muzzle contacts compared to knowledgable females and juveniles, while also being less likely to be initiators."48: What is meant by "potential" novel foods?

Anything an animal *might* try to eat to see if it is edible. To consider something to be a novel food, to me, it should first of all actually be edible, and in this context, it should be investigated for it’s potential as a food by the animal. Not everything novel in an animal’s environment is a novel food. This is why I used the word “potential”. I would prefer to leave it like this as I think it makes sense, and it did not raise concern in previous versions. If the editor has the same issue and wants us to remove the term ‘potential’ we will do so.

50-51: the line between "obtaining novel information" and "produces information or knowledge, potentially facilitating information" is murky. New information gained by an individual isn't always reduced by them…perhaps something like "and has the potential to produce information on which other group members can act." instead.

We believe the reviewer refers to this line “…directly from the environment requires overcoming neophobia and engaging in exploration, tendencies for which may vary between individuals”? We are not sure why this is unclear. We are simply saying that exploration tendencies are variable between individuals.

We do not understand the reviewers comment: “New information gained by an individual isn't always reduced by them” – our best guess is that you mean the information is not always “used?

Here we are not yet talking about information being used by other individuals, nor even being used by the individual finding the information – this is why we said it “potentially” facilitates innovation, not that it always leads directly to it. We have broken this sentence into two and hope it is clearer now: “Obtaining novel information directly from the environment requires overcoming neophobia and engaging in exploration, tendencies for which may vary between individuals [5]. If this information is used, it can facilitate innovation.” (L51-53)

58: Behavioural patterns don't have be novel to be adaptive in new environments, as in the definition of innovation provides above-innovation could be a solution to a novel problem, including generalizing one behaviour, e.g., extractive foraging like getting acacia seeds from seed pods, to a novel situation, like extracting peanuts.

Yes true, this sentence did not come across as intended – we meant that the behaviour in interaction with the environment would be novel, whether it was a novel behaviour or not, as per the definition in the previous sentence, but indeed this is not how it sounds. We have updated it now to “Individuals must interact with the environment in novel ways, either using novel behaviours with known environmental features, or performing familiar behaviours on novel aspects of the environment, which additionally requires behavioural plasticity [11]” (L 59-62).

60: What's "this"? Behavioural plasticity? Behavioural patterns?

Yes – we have now clarified this in the text (L 62-63).

61: The risks association with novelty and innovation are unclear here. Up until this point, innovation and novelty have been largely been framed as positive, whereas only neophobia was mentioned with avoiding ris

We gave examples of risks associated with novelty at the start of this paragraph and have added to this sentence now so hope this is clearer (L 46-50)

66: Are juveniles required to innovate to learn about their environments? Learning about what is available as food in your group is not the same as finding a new food. Similarly, do dispersing animals need to innovate, or do they need to use previously-learned social skills to ingratiate themselves to a new group? Neither of these examples are a "a solution to a novel problem" or a "novel solution to an old one". Learning what to eat is not a novel problem…much like learning who your allies are isn't a novel one either. What really is the problem here is whether innovation happens at the level of the individual or group… certainly learning what to eat as a juvenile, or integrating into a new group, isn't a novel problem for this species, but one every animal must meet (save females re: dispersal). What I mean to say here, just because juveniles might be more prone to innovate, it's not necessarily from necessity, but could be a result of a developmental period that functions primarily to allow them to learn about their environments-while behavioural flexibility is necessary for innovation, it is not the same as innovation.“Are juveniles required to innovate to learn about their environments? “

No, we have updated the sentence to read now: “For example, innovation might be more likely in juveniles who may be less neophobic due to their need to learn about their environment before adulthood…” L 68-70

*“do dispersing animals need to innovate,”* – they have to explore and approach and interact with novelty, which may well increase the likelihood of innovation, which can also be due a transitory exploratory behavioural syndrome as specified in the introduction (L 71-72) and in the discussion (L 315-316).

“*Learning what to eat is not a novel problem*” – not if they learn from someone else, but if they individually learn to eat a novel food we would count this as an innovation. This is explicitly stated in the definition of innovation that we provide (“a new ecological discovery such as a food item not previously part of the group“, L 54-55). It also relates to the earlier part of that definition, i.e. applying an old solution (eating) to a novel problem (a novel object that was not previously known to be edible).

70: Why "nonetheless" here? The points following "nonetheless" seem to follow the points before it, and are not in contrast.

We have changed it to “on the other hand” (L77)

77: How is the risk diminished? Are captive animals are less neophobic because they are fed-is risk assessment, for which neophobia is a conserved trait across many species, is ontogenetically determined? Or is the point here that is more difficult to study such phenomenon in captivity, because there is no risk? I assume it's this latter point, but such risk assessment is never addressed in the current study, other than to mention that wild animals often experience changing environments, especially resulting from anthropogenic origins.

We are simply trying to point out that the risk of ingesting toxins is diminished in captivity due to being fed with edible food only. We have added a reference to support this wild-captive difference [L83-87].

77-79: Individual differences with regard to innovation and behavioural plasticity has been shown to be true across many studies, including vervets, though there has been some conflicting evidence (cited below)…would it be better to say more work is needed, given the conflicting evidence?Bono, A. E. J., Whiten, A., Schaik, C. V., Krützen, M., EichenBerger, F., Schnider, A., and van de Waal, E. (2018). Payoff-and sex-biased social learning interact in a wild primate population. Current Biology. Current Biology, 28(17), 2800-2805. https://doi.org/10.1016/j.cub.2018.06.015Bono, A. E. J., Whiten, A., Schaik, C. V., Krützen, M., EichenBerger, F., Schnider, A., and van de Waal, E. (2018). Payoff-and sex-biased social learning interact in a wild primate population. Current Biology. Current Biology, 28(17), 2800-2805. https://doi.org/10.1016/j.cub.2018.06.015Renevey, N., Bshary, R., and van de Waal, E. (2013). Philopatric vervet monkey females are the focus of social attention rather independently of rank. Behaviour. Behaviour, 150(6), 599-615. https://doi.org/10.1163/1568539X-00003072Canteloup, C., Hoppitt, W., and van de Waal, E. (2020). Wild primates copy higher-ranked individuals in a social transmission experiment. Nat Commun. Nat Commun, 11(1), 459-469. https://doi.org/10.1038/s41467-019-14209-8

These cited studies do not represent conflicting evidence neither of innovation nor on social learning biases as we already offered some explanation regarding results’ inconsistencies. Bono et al. (2018) study trained models to use one technique over the other, while Canteloup et al. (2020) did not train any models, then the ‘innovators’ are not comparable in those studies. As one can read in Canteloup et al. (2020) p 6:” While previous studies of vervet monkeys reported both a female bias (van de Waal et al. 2010; Bono et al. 2018) and a mother bias (van de Waal et al. 2012; 2014; 2013) our findings did not identify any such biases. On the one hand, such inconsistencies could be due to our relatively small sample size compared to previous studies that tested more than two groups. On the other hand, discrepancies might be explained by the fact that in both cited studies (van de Waal et al. 2010; Bono et al. 2018), female and male models were of high social rank while in our study, they were of varying social ranks. The possibility that different results could have arisen by running the same kind of experiment with only low rankers or low-ranking females and high-ranking males as models is an open question. Finally, in the above cited studies (van de Waal et al. 2014; 2013), only infants of less than one year of age were tested whereas in our study, individuals of all ages took part in the experiment. It is then possible that young infants focus on maternal figures during a first phase of learning and later widen their attention during a second phase of learning, focusing on specific individuals such as high-rankers (Whiten and van de Waal 2018) who could be considered as experts (Whiten and van de Waal 2018; Kendal et al. 2015)”. That being said, we have added that “…more worked is needed on the topic.” (L 89).

79: "For example" might work better here rather than "moreover", since it's continuing the previous point. The reference to chimpanzees explicitly here is unnecessary, as the citations used include species beyond chimpanzees. "For example, across many species where males disperse, dispersing individuals…" would work better.

We prefer to keep “moreover” to emphasise the shift of focus towards dispersers whilst continuing the overall point. (“Moreover” can be used to continue a point – similar to “furthermore”; L 89).

80: Citations [19] and [20] here use capuchins, not chimpanzees

We have updated this citation to include all the subsequent references related to this point. (L93-95)

85: Add "While these studies show that dispersing individuals…experimental (no "but")". However, at this point it is not clear to me why we need to compare multiple groups experimentally-this needs support.

Good suggestion, thanks. (L96)

However, at this point it is not clear to me why we need to compare multiple groups experimentally-this needs support.

To look at between-group transmission experimentally, multiple groups are required by definition, and the more groups the better as results will be more representative of the population. In addition, as dispersers represent few individuals, decent sample sizes can be built up if multiple groups are studied. We have added “resulting in a small pool of evidence” to reflect this. (L98)

70-100: This paragraph is very confusing to follow, as there are multiple independent points being made, including how social learning is beneficial, how the dispersing sex can import innovations or create them, a brief mention of the interface between social learning and innovation (i.e., it is implied that they are separate processes, but all of the benefits of this introduction point to species-level benefits of innovation, which require innovations to spread, so the brief mention of social learning here seems too minimal), and a discussion of social learning modalities.

We have split it into two paragraphs, to separate the social learning modalities from the rest. (second paragraph starting L99)

97: Why the mention of social tolerance here? It is not clear how social tolerance speaks to the questions asked by this study.

We are referring here to Nord et al. (31) study who suggest that social tolerance may constrain information transmission with tolerant individuals accepting more individuals in close proximity than more intolerant ones would; and some individuals are tolerated more than others are. To make this more obvious, we have added: “which can be affected by age, sex and rank” (L107-109) which are all parameters that we measure and analyse in this study.

103: What's "this"?

“This” referred to the sentence directly before the word “this”. It is now updated to be more specific (“Their diverse habitats, including those highly modified by humans,…” (L 115)).

107: I'm still not sure why males need to innovate? Don't they, at most, need to generalize the behaviours of their previous groups to a new one? Why must they innovate?

The point is that they might be more predisposed to innovate because they have to seek novelty (new group, new habitat, interact with new group mates using familiar behaviours with new individuals, explore an unfamiliar area…). Thus, relative to females, the likelihood of them ending up innovating is increased, even if they don’t have to be overall extremely innovative – just more likely to than the philopatric females.

We have expanded L 118 to hopefully make this clearer: “Frequently dispersing males may serve as vectors of information between groups. In addition, if dispersal triggers an increase in exploratory behaviour, necessary to seek novelty in order to leave one group to join a new one, around the dispersal period they may also become more likely innovators in novel environments [12], potentially facilitating behavioural adaptation to diverse habitats across their geographical range [33].” (L 118-122).

123-124: Consider changing to "Our observations of innovation are limited in number, but further testing of the hypotheses we propose, as a result of our exploratory analysis that we present here, may aid our understanding of animal innovation.

Thanks, we have updated this sentence as such (now L 137-139)

125: Consider changing to "Given that animals learned to eat a novel food source, a behaviour that spread socially [24], (1a)…"

We disagree here with this change as when we asked (1a) who innovated, this was the first exposure of animals to the novel source so they did not already learn to eat it.

126-129: Why did you expect this?

We expected it because adults are more likely to be preferred learning models than juveniles in agreement with the three phases in the ontogeny of social learning in primates (L 141-145).

130: "which" implies the results were known beforehand; "whether" might work better here.

This has been updated (L 146).

131: Change "over all" to "overall"

In this instance “over all” was actually correct – we were using the two separate words as separate words, but have changed it to “across all” to be clearer (L 147-148).

131-133: Why differentiate across exposures here, especially since the predictions are the same? I know there are 2 different models, but I need to know why here so that I can understand the differing predictions. Moving the explanation as to why the first 4 exposures were considered to here would be helpful.

We have removed the differentiation between the first and across all four exposures from here, since the predictions are anyway the same (L148-150). We chose to do this instead of moving the explanation here as the latter seems more complicated and will disrupt the flow of the predictions if we were to start explaining the methodical decisions here.

132: What's "this"?

We changed “this” to “adoption of the innovation” (L 148).

134-136: There seems to be a bit of double-dipping here, as the findings from one dataset (at least for 2 groups) are used as evidence for a prediction for data in the same dataset (the current study). Additionally [24] and [39] found that higher-rankers are more likely to be observed, not that they were more likely to uptake a novel food. In fact, the common prediction here is that higher-ranking animals should be more neophobic, because they have better access to food and thus eating unknown food is riskier for them given their prime access to food overall:Wolf, M., van Doorn, G. S., Leimar, O., and Weissing, F. J. (2007). Life-history trade-offs favour the evolution of animal personalities. Nature. Nature, 447(7144), 581-584. https://doi.org/10.1038/nature05835Greenberg, R. (2003). The Role of Neophobia and Neophilia in the Development of Innovative Behaviour of Birds Animal Innovation. In S. M. Reader and K. N. Laland (Eds.), Animal Innovation (pp. 175-196). Oxford University Press. https://doi.org/10.1093/acprof:oso/9780198526223.003.0008Laland, K. N., and Reader, S. M. (1999). Foraging innovation in the guppy. Animal Behaviour. Animal Behaviour, 57(2), 331-340. https://doi.org/10.1006/anbe.1998.0967But see:Amici, F., Widdig, A., MacIntosh, A. J. J., Francés, V. B., Castellano-Navarro, A., Caicoya, A. L., Karimullah, K., Maulany, R. I., Ngakan, P. O., and Hamzah, A. S. (2020). Dominance style only partially predicts differences in neophobia and social tolerance over food in four macaque species. Scientific reports. Scientific reports, 10(1), 1-10. https://doi.org/10.1038/s41598-020-79246-6Drea, C. M. (1998). Social context affects how rhesus monkeys explore their environment. American journal of primatology. American journal of primatology, 44(3), 205-214. https://doi.org/10.1002/(SICI)1098-2345(1998)44:3%3C205::AID-AJP3%3E3.0.CO;2-%23

As the reviewer highlighted, predictions about the effect of rank on uptake of a novel food in the literature are various. We analysed here rank in the models because it was likely to have an effect as it was found to have an effect in the Canteloup et al. 2020 paper with the same peanut experiments on two groups, and we wanted to control for this, and also to explore what might be the influence of rank in this context.

140-142: This makes sense to me, but I have no idea why this prediction is made. Perhaps moving the explanation given to the me

The reviewer’s comment is incomplete, and it is not clear what changes they are requesting about.

146-147: "the media" implies that this is how animals are learning to eat the peanuts…but the author replies to reviewers mention multiple times that this is not what is meant.

According to the Merriam-Webster online dictionary, a medium is “a means of effecting or conveying something such as a channel or system of communication, information, or entertainment”. Following this definition that does not imply any mechanism of learning (how animals are learning), we changed the sentence for: “muzzle contact being a medium to obtain information about what conspecifics are eating” (L 160-163).

We are suggesting that the monkeys are using the muzzle contacts around the novel food to obtain information about it, thus indeed potentially informing whether they eat the peanuts or not – as the reviewer mentions here. But we are focussing on whether we have evidence that muzzle contacts convey information about the novel food, not whether and how this information is used. We are not looking at whether the monkeys eat peanuts *because* they’ve done muzzle contacts, but rather at whether their MC behaviour is influenced by whether they have eaten or not (the opposite direction of causality, not that we can measure the causality directly, but rather we believe we build evidence in this paper for the latter). As we have said before, there may be a lot of other information that is important too in the novel food-learning process, such as visual cues (e.g. see Canteloup et al. 2021 on the opening techniques which follow an observation network) and individual learning that might be occurring in the process of learning to eat peanuts that we are not interested in in this paper. We hope this is clearer now.

147-150: This has been previously found in [31]; thus there is both theoretical and empirical support for this prediction.

Reference [35] was indeed cited at this place but we added another reference to that study (L 166).

152: [31] found this, and hypothesized that social tolerance is necessary for muzzle contact to afford foraging information, but perhaps and additional citation here about how lower ranking animals are tolerated by fewer group members would help make the point.

This is already clearly stated L169-171.

153-154: Initiated vs targeted…via muzzle contact?

We remove this whole paragraph as we removed model 6 from the paper.

158-161: I don't follow…this seems to assume that initiators are seeking information and this seeking will outweigh any social tolerance constraint, but only previous study of muzzle in vervets found that tolerance was the best predictor of the behaviour. Thus, this prediction needs more support as to why it is in the opposite direction of what the literature shows, i.e., that social tolerance constrains information-seeking and information spread, akin to Carter's (2016) sequential social learning hypothesis.

This whole paragraph has been removed now.

161-173: What kind of different experiences of novelty arise from the life history trajectories of the philopatric vs. dispersing sex? Again, why do dispersing animals experience more novelty? Why should we expect the groups to which they are dispersing to have significantly different diets that we can call "novel"? Do dispersing animals need to gain totally new information? Surely not, as the kinds of foods available are likely very similar and behavioural generalization can do a lot of work. When it comes to conspecifics, isn't a plausible alternative hypothesis that dispersing animals need to enter groups using the same skills needed to integrate in to the adult social networks as they age in their natal groups before dispersal…so what counts as novel here? Again, I see these problems as being neither novel or the success of dispersing animals after integration into new groups as being dependent on a novel solution. It seems to be the novelty of interest here is much larger, as mentioned at the beginning with reference to anthropogenic-induced changing environments, rather than the kinds of problems these animals have encountered throughout their evolution.

This whole paragraph has been removed now.

164-165: Why this prediction? I can think of some reasons why this is, but there is no support for this prediction that naive adults should initiate and receive at the same rates…prior evidence [31] suggests that adults should more often be targeted than initiate, so it seems here that this prediction relies on explicit knowledge seeking, which requires the prediction the muzzle contact is primarily used to gain novel information.

This whole paragraph has been removed now.

166-167: Why should females initiate if they don't "need" info, which is what this prediction is implying…presented like this, this prediction reads as if the results were already known when it was made.

This whole paragraph has been removed now.

169-171: Why would malesy stop initiating? Why does initiating influence receiving? One does not preclude the other…

This whole paragraph has been removed now.

170-173: I don't follow this prediction at all… that knowledgeable males are somehow more tolerated…doesn't the work reviewed in the introduction at least imply that new immigrants have novel information, and by definition, new immigrants are less known to the group, so should be interacted with less. How does a muzzle contact initiator know that a new male is knowledgeable? And why wouldn't a new male be less tolerated by others compared to an established male, who has relationships with the group? Again, I'm not sure of the dismissal of tolerance here, when the only previous work on muzzle contact in vervets found that social tolerance, above all else, influences muzzle contact behaviour? Especially given that prediction 2b makes a social tolerance prediction, that lower ranking animals will initiate less than higher ranking animals.

This whole paragraph has been removed now.

179-182: Why mention this here? Perhaps this would be better at the very beginning of the results, or near where the differentiation is first referred to in the results.

It was placed there at the suggestion of a reviewer. We moved this paragraph back to the beginning of the Results section now (L 174-177).

209: Should this be "his" instead of "their"?

We changed it for “the group’s” as it refers to the “group” (L 190).

213-221: This use of uptake is confusing… in the reply, the authors state " the reason we talk about 'uptake' rather than social learning is that we really see this as a case of social disinhibition of neophobia, rather than more detailed social learning such as copying or imitation" but this disinhibition hypothesis is never mentioned in the introduction.

In the introduction, we talk about neophobia being reduced after seeing conspecifics eat a novel food in various species L 79-80. We modified the wording of our prediction for this result to be more explicit about this: “We expected, when the innovators or initiators (in case of immigrant males importing the innovation) were adults rather than juveniles or infants, greater neophobia reduction and therefore faster and more widespread uptake of the novel food…” – L 141-144

The introduction needs to make clear this distinction, and why, despite that this behaviour was previously shown to be socially transmitted, social learning language here. I see no reason not to report that this behaviour is socially transmitted and that this study takes the opportunity to explore who innovated and whether socio-demographic variation corresponded with innovation, as well as the opportunity to further explore muzzle contact as a means of learning about novel foods given previous evidence showing that muzzle has the potential for being a learning modality, rather than proposing an entirely different mechanism.Also, how does one prove the difference between the uptake of the innovation being the result of social disinhibition and the topography of opening the peanut being socially transmitted? I understand the use of EWA to show the latter, but am not sure how that is separate in fact from the former…how does one show the approach and willingness to interact is only socially facilitated, but the opening itself is socially learned? Especially given that all of the results in this study are presented in regard to who extracted and ate the peanuts, and not some other measure of neophobia.

To clarify: The first paragraph of results of the section ‘(1a) Who innovated and how did it affect the extent to which the innovation was adopted by the group?’ (L189) refers to the individuals who innovated – ate the novel food first. Innovation relies on individual learning as one individual has to start eating the food the first time. The paragraphs of this section ‘(1b) Socio-demographic variation in learning the innovation’ (L213-228) refers to who ate the food over four exposures, and here social learning might happen as shown in Canteloup et al. (2021) but, as one can read in that paper, a combination of individual and social learning is at stake when learning which peanuts opening technique to use. In that sense, we changed “uptake” for “learning” (L 212). We hope that this explicit statement clarifies things. If not, we are of course willing to modify the term if requested by the editors.

Reviewer #2 (Recommendations for the authors):I commend the authors for their hard work in improving their manuscript to accommodate the comments raised by myself and the other reviewers. However, I still feel there is considerable conceptual fuzziness that constrains a clear interpretation of the data presented here, as well as some remaining issues with the analysis. Much of this is made apparent in the authors' Reply to Review, so I will primarily address this. Below that, I have some more minor comments on the revised manuscript.1) Conceptual and inferential ambiguity"My comment: Line 281: More detail needed. Did these knowledgeable individuals typically have their mouths full of the target food during these events? If so then it seems parsimonious to assume the muzzlers were simply following this rather than tracking knowledge-states.Authors reply: We do not claim that they track knowledge states – we are claiming that they can tell who is currently eating or has eaten a food that they do not know about, and try to obtain information about that food. We use the word "knowledgeable" for our human readers to easily identify and refer to "individuals that have already learned to extract and eat peanuts". We never report in the manuscript that we are inferring that the monkeys track the knowledge state. We do assume that if they are close enough to muzzle contact, they are close enough to have probably seen them eat the food.""…we never report in the manuscript that we are inferring that the monkeys track the knowledge state." Throughout the manuscript the authors make statements to this effect…"I'm particularly surprised by this final comment since one need not even read past the abstract to see that it is clearly untrue: "Finally, knowledge influenced females and juveniles less than males and adults in becoming more likely targets than initiators.". The manuscript is riddled throughout with examples of such causal language that heavily implies a direct effect of knowledge on the outcome measures. This is extremely misleading and serves no purpose. The word 'knowledge' should be removed from the manuscript entirely and the authors find another way to describe their variable. For example, why not just call the 'knowledgeable' individuals "demonstrators"?Below I answer several comments at once:"We did not intend to claim that muzzle contact was the specific mechanism by which individuals learned to extract and eat peanuts – we rather use this experiment to evaluate the function of muzzle contact in the presence of a novel food.""For this, and the above points: We did not record an observation network for the groups added in this study and are not able to answer this – it is not the focus of this study. For this reason, we do not make claims in this line in the present study, and are cautious with our social learning related language. Whilst we examine the role of muzzle contact in acquiring information about a novel food, we do not expect this behaviour to be a necessary prerequisite in being able to extract and eat this food – indeed many individuals who learned to eat did not perform muzzle contacts. This aspect of the study is about using this novel food situation to explore whether muzzle contact serves information acquisition – which our evidence suggests it does. Moreover, the processing of this food is not complex and is similar to natural foods in their environment, and we do expect individuals to be capable of reinventing it easily (and this point with Tennie's hypothesis is actually discussed in Canteloup et al. 2021 paper) – but the point here is that their natural tendency is to be neophobic to unknown food, and therefore they do not readily eat it until they see a conspecific doing so, after which they do. And we also used this opportunity, though in a very small sample size, to investigate which individuals would overcome that neophobia and be the first to eat successfully.""See above – the reason we talk about 'uptake' rather than social learning is that we really see this as a case of social disinhibition of neophobia, rather than more detailed social learning such as copying or imitation, as it would be in a tool-use setting, for example (though in Canteloup et al. 2021 paper, evidence is found that the specific methods to open peanuts are socially transmitted).""…there is a distinction between information acquisition and information use – obtaining olfactory information about a novel resource that conspecifics are eating is not the same as learning a complex tool use behaviour for which detailed observation of a model is required. We are not claiming that muzzle contact is THE mechanism by which the monkeys learn how to eat the food"To summarise: When I suggested the authors have implied a role in social learning, they deny this (okay! But I'm unsure about the need for evasiveness on this one – there are more kinds of social learning than just action-copying). Nevertheless, they argue that the monkey are 'gaining information' about the food and that the decline in MC as they become more knowledgeable implies a role in learning (social or asocial) or 'overcoming neophobia'. This seems plausible and a worthy hypothesis to test!However, when I asked for evidence that individuals who MC more often are more likely to learn how to eat the food, the authors refused to examine this on the basis that "MC is not THE mechanism by which learning occurs". Regardless of whether it is THE mechanism, or simply a means of overcoming neophobia, if MC serves the function the authors have argued then it should lead to an increase in the likelihood or rate of uptake – otherwise what is the point? The authors refusal to support their argument with easily accessible data (they have apparently already recorded the identity of all individuals and their feeding/Mc behaviour) that would robustly confirm the behavioural function one way or the other is quite frustrating.

Whilst we have all of the muzzle contact interactions coded with the identities of individuals, and we know which exposure individuals successfully shelled and ate their first peanut, we do not have the exact timing of the latter for all individuals. We therefore cannot do the analysis that the reviewer proposes without extensive recoding of videos and the authors responsible for coding the videos with individuals’ identities are no longer employed in this field of work to do so.

In addition, we stand by our points, that information acquisition and information use are not the same, and that MC is not the only way to gain information. Specifically, not all monkeys that started to eat peanuts engaged in muzzle contact beforehand – this is our point about it not being the only way to obtain information – less tolerated individuals are unlikely to be able to engage in MC as easily (see Nord et al.), and therefore must rely on other kinds of information. Thus, we focussed at the group level on whether muzzle contacts decreased as the group increasingly ate peanuts.

Nonetheless, we do now report a different analysis that illustrates that as more individuals gain knowledge of the food, muzzle contact rate decreases.

In fact, the authors do present some data that contradicts their hypothesis:Line 681: "Inspection of Figures 4A and 4D suggests that juveniles, relative to adults, still initiate more than they are targeted even when knowledgeable."Why should knowledgeable individuals muzzle-contact at all? These individuals already have the information they need. This is a major hole in the authors' argument.

Even if muzzle contact is used, particularly, to gain information about unknown food, an uncertainty might remain. Moreover, primates are social animals, and we cannot exclude a social function to this behaviour.

"We recorded muzzle contacts visible within 2m of the box, so individuals were not necessarily eating at the box at the time of engaging in muzzle contacts. However, the majority of muzzle contacts that we could record took place directly at the edge of the box – at the location where the food is accessed – so an individual would not likely be if they were not able to have access to the food. It is possible they could be there and not eating, but they would not have been chased off, otherwise they would not be able to engage in muzzle contacts there. But it is not entirely clear what the reviewer's point is here."If muzzle contact was only recorded within 2m of the food source, is it any wonder that knowledgeable individuals were chosen more often? Surely the majority of individuals at the food are those who have figured out how to eat it. See the comment below this one.

Not necessarily as all the individuals that came within 2m of the box did not eat the peanuts, especially at the beginning when most muzzle contacts happened. In addition, individuals that had not yet eaten would have to approach the box to begin eating too. We understand your logic here, but we disagree with it because there being more knowledgeable individuals at the box already does not preclude those knowledgeable individuals to do MCs towards naïve individuals that approach the box. So even around the experimental setup muzzle contact could theoretically be initiated by both naïve and knowledgeable monkeys.

"My comment: What proportion of PRESENT (not total) individuals were naïve and knowledgeable in each group for each trial (if 90% present were knowledgeable, then it is not surprising that they would be targeted more often)?Authors reply: We agree somewhat with this statement, but given the multiple ways we show the effect of knowledge – both at the individual level and the group level (effect of exposure number i.e. overall group familiarity) – we feel we present enough evidence to establish the link between knowledge of the food and muzzle contacts. We find that the model showing the interaction between exposure number and number of monkeys eating on the overall rate of muzzle contacts actually addresses this issue, because we see that when many monkeys are eating during later exposures when many were indeed knowledgeable, the rate of muzzle contacts is massively decreased. Moreover, if 90% of the individuals present are knowledgeable, then only 10% of the individuals present are naïve, and we show both that knowledgeable individuals are targeted, but also that naïve individuals are initiators."The authors have not really addressed my original point here, so I apologise if it was unclear. First, I accept the authors' conclusion that knowledgeable individuals are less likely to carry out a MC (but see below for problems regarding their interpretation of this). Instead, I was raising a point of basic sampling bias and statistical inference: If the majority of individuals at a feeding site are knowledgeable, then even a blindfolded individual who is choosing recipients are absolute random will select knowledgeable individuals more frequently. If all of the knowledgeable individuals are male, a blindfolded individual will similarly demonstrate a "bias" towards male, knowledgeable individuals. If this is not factored into the analysis then it is not inferentially sound.

Thank you for clarifying this – we do see and agree with your point here. However, our data do not fit the pattern of bias that you describe. We have a situation where when very few individuals were knowledgeable, and many were naïve, the knowledgeable minority were targeted by the naïve majority, and this decreased as more individuals became knowledgeable. Given that we have an increase of knowledgeable individuals over time AND a decrease in muzzle contacts over time, we think this counteracts the random bias that was pointed out here. In addition, we can see in Figure 3A, that at the beginning of the experiment when very few individuals were eating, there were the highest rates of MC, which decreased across the experiment.

"…but we do believe that the clear separation between naïve individuals initiating and knowledgeable individuals being target, and the decrease of the rate of this behaviour as groups' familiarity with the food increases – is good evidence that this behaviour functions to acquire information about a novel food."That is one interpretation (but see comment above re: sampling bias for initiators) – Another explanation is that these behaviours are simply mutually exclusive at a given moment in time: once they know how to eat the food, they prefer to spend their time doing this than engaging in MC behaviour. Rates of resting, grooming, etc within 2m of the food presumably also decrease once the monkeys have figured out how to eat it, not because there is any causal relationship between these behaviours but because they can only do one thing at a time and feeding is a priority.

We understand the point here (essentially an activity budget point) and the comparison of MC rates to rates of other behaviours such as resting and grooming in the context of consuming food. However, we did not observe particularly high rates of individuals coming to the vicinity of the food and engaging in resting or grooming at that location, whether the food was novel or not. This contrasts with what we observed of MC – which became unusually high, in the vicinity of the box, when the food was novel, decreasing as the food became familiar. We believe this difference does suggest a causal relationship of the novelty of the food and conspecifics eating it as triggering the MCs.

During experiments with food rewards, individuals tend to not rest or groom at the vicinity of the food box. Muzzle contacts are being carried out specifically at this location where their group mates are consuming a novel food. We do also see individuals follow other who carry food away from the box, and MC them at the location where they stop to eat. Unfortunately, we did not collect data in this experiment on occurrence of grooming and resting at the food box that we can analyse to show statistically that grooming and resting rates are not increased relative to baseline levels in the vicinity of the box. We believe that the high rates of muzzle contact that we see directly at the vicinity of the box do imply a causal relationship that the monkeys have come to this place to do muzzle contacts prior to engage with the novel food.

2) AnalysisThe authors have heavily revised their original analysis and it is largely improved. I have a few remaining issues which I describe below."My comment: The text for this muzzle-contact analysis would indicate that this model was not fit with any random effects, which would be extremely concerning. However, having checked the R code which the authors provided, I see that Individual has been fit as a random effect. This should be mentioned in the manuscript. I would also strongly recommend fitting Group (it was an RE in the previous models, oddly) and potentially exposure number as well.Author reply: The model about muzzle contact rate never contained individual as a random effect because individuals are not relevant in this model – it is the number of muzzle contacts occurring during each exposure. However, the reviewer might refer here to the model that we forgot to provide the script for. Nonetheless, we have substantially revised this model, it now (Model 3) includes all groups, and has group as a random effect."I do not accept that individual is not a relevant random effect. I understand that the model is intended to examine group-level rates of M-C, but groups are made of individuals. Let us imagine a scenario where a single individual is a highly prolific muzzle-contacter in group BD, accounting for 95% of M-C events, and NH contains no such individuals. An analysis that takes a straightforward group rate without accounting for individual contributions will likely find a significant difference between the two, driven by a single individual. If the authors have structured their data and analysis in such a way that they cannot control for this factor then that is an issue. One "quick and dirty" solution, that would require a minimal amount of restructuring of the data, would be to take an individual rate for each monkey in a group, or at the feeding site, or whatever, and then derive the group average from this. Otherwise, it is not clear what we can infer from this analysis.

We have re-structured the data and redone this analysis with both individual and group as random effects and hope the reviewer find these changes satisfactory. Please see L 612 in the methods, as well as Table 3 for updated results.

"Authors: We have now checked for overfitting in our models."Where is the evidence of this, please? There are metrics and methods that can be used to achieve this (such as AIC/LOO-based model comparison approaches I suggested in my last review) but the authors do not report them.

We calculated the variance inflation factor (VIF) for each variable in all models and, except for the variable “number of monkeys eating” in model 3 where VIF = 4.36, suggesting a moderate correlation, all VIF are <3, meaning that there is no overfitting. Moreover, we compared the AIC of our models including random effects with AIC of models without random effects, and the lowest AIC were always those of the models including random effects. This is now specified in the method (L 650-654) and in the annotated R script.

"We included individual as a random effect, but we did not include group as a random effect here for two reasons. First, we did not have any theoretical basis to expect residing in different groups to have an effect here, since we were concerned with the effects of life history strategies of individuals on their information acquisition behaviour, which should not differ for individuals from different groups."This is not theoretically sound. Individuals from groups are more likely to be similar than individuals from different groups – this is the purpose of grouping variables. They live in similar ecologies, share life history events, and are more closely related.

We now removed this model so this is no longer an issue.

[Editors' note: further revisions were suggested prior to acceptance, as described below.]

Thank you for resubmitting your work entitled "Role of immigrant males and muzzle contacts in the uptake of a novel food by wild vervet monkeys" for further consideration by eLife. Your revised article has been evaluated by George Perry (Senior Editor) and a Reviewing Editor.The edits to the manuscript were much appreciated but unfortunately have also brought to our attention some additional issues with your statistical analysis that must be addressed, as outlined below.1. The issue is that once you reported your dispersion parameter results, it is now clear that Models 4 and 5 are highly underdispersed, and model 3 moderately so. Underdispersion can be considered as much an issue as overdispersion for poisson models so we urge you to rethink the error structure used for these models so that you do not violate the assumptions of a poisson distribution.

We consulted a statistician from the University of Lausanne for the issue 1 listed here below. He confirmed to us that our dispersion test were correct and that our models are not over or underdispersed, but that what we reported as dipsersion parameter is not that. Here his feedback:

'The testDispersion() function in R is actually not very useful, because it pretends to display the dispersion (by writing "dispersion = 0.0017655, p-value = 0.592"), but actually, it is not.

The value (0.0017655) is actually saved in the object under the name "statistic"; so it is not the dispersion, but a ratio that is calculated by the model by comparing the observed model to simulated data. This ratio is then compared to 1, but there is no intrinsic notion of scale (e.g. the fact that the observed value is 0.0017 does not mean much, and certainly not that the dispersion is so low; the simulation are then run to estimate the significance of this difference).

In this case, p=0.59, so that the value is not significantly different from 1, and so there is no evidence that the data is significantly underdispersed.'